# Factors Controlling Hydrothermal Nickel and Cobalt Mineralization—Some Suggestions from Historical Ore Deposits in Italy

**Marilena Moroni [1,\*], Piergiorgio Rossetti [2,\*] , Stefano Naitza [3,\*], Lorenzo Magnani [1], Giovanni Ruggieri [4] , Andrea Aquino [5], Paola Tartarotti [1] , Andrea Franklin [1], Elena Ferrari [1], Daniele Castelli [2], Giacomo Oggiano [6] and Francesco Secchi [6]**

1   Dipartimento di Scienze della Terra, Università degli Studi di Milano, 20133 Milano, Italy
2   Dipartimento di Scienze della Terra, Università degli Studi di Torino, 10125 Torino, Italy
3   Dipartimento di Scienze Chimiche e Geologiche, Università degli Studi di Cagliari, 09124 Cagliari, Italy
4   IGG-CNR, UOS Firenze, 50121 Firenze, Italy
5   Dipartimento di Scienze della Terra, Università degli Studi di Firenze, 50121 Firenze, Italy
6   Dipartimento di Chimica e Farmacia, Università degli Studi di Sassari, 07100 Sassari, Italy
\*   Correspondence: marilena.moroni@unimi.it (M.M.); piergiorgio.rossetti@unito.it (P.R.); snaitza@unica.it (S.N.)

**Abstract:** We compare three poorly known, historical Ni–Co-bearing hydrothermal deposits in different geological settings in Italy: The Ni–Co–As–Sb–Au-bearing Arburese vein system (SW Sardinia), the Co–Ni–As-rich Usseglio vein system (Piedmont), and the small Cu–Ag–Co–Ni–Pb–Te–Se stockwork at Piazza (Liguria). These deposits share various (mineralogical, chemical, thermal, and stable isotopic) similarities to the Five Element Vein-type ores but only the first two were economic for Co–Ni. The Sardinian Ni-rich veins occur in Paleozoic basement near two Variscan plutons. Like the Co-rich Usseglio vein system, the uneconomic Piazza deposit is hosted in an ophiolite setting anomalous for Co. The Sardinian and Usseglio deposits share a polyphasic assemblage with Ni–Co–As–Sb–Bi followed by Ag-base metal sulfides, in siderite-rich gangue, whereas Piazza shows As-free, Ag–Pb–Te–Se-bearing Co–Ni–Cu sulfides, in prehnite–chlorite gangue. Fluid inclusions indicated Co–Ni arsenide precipitation at ≈170 °C for Usseglio, whereas for the Sardinian system late sulfide deposition occurred within the 52–126 °C range. Ore fluids in both systems are NaCl-CaCl2-bearing basinal brines. The chlorite geothermometer at Piazza provides the range of 200–280 °C for ore deposition from $CO_2$-poor fluids. Enrichments in Se and negative $\delta^{13}C$ in carbonates suggest interaction with carbonaceous shales. These deposits involve issues about source rocks, controls on Co/Ni and possible role of arsenic and carbonate components towards economic mineralization.

**Keywords:** Ni–Co arsenides; five element vein-type deposits; fluid inclusions; basinal brines; carbonates; chlorite geothermometer

## 1. Introduction

Nickel-and cobalt-rich hydrothermal deposits are interesting both for the economic importance of their metals (especially highly sought-after cobalt) and for aspects of their genesis still not completely understood. The detailed studies of the impressive, early works on these deposits, dating back to the 1980s and summarized in [1], outlined several important features including the low-temperature characters of this type of mineralization and its tendency to regional-scale development. The available, limited yet precious experimental data on Ni and Co in hydrothermal conditions, e.g., [2–5], have

highlighted a different behavior of these metals in solution and the need of further research for unraveling poorly understood aspects. However, recent investigations on natural ore deposits and related modeling, e.g., [6–8], have provided clues about the components of the mineralizing fluids as well as on reaction mechanisms effective for metal deposition. Yet speculations still exist on various issues, including the possible mechanisms of selective enrichment of Co versus Ni and the sources of Co and Ni, which may be multiple and diverse, or exotic, even in fluid-related ore deposits hosted in mafic-ultramafic rocks, as discussed in [9].

In this work we try to contribute to the discussion by presenting some preliminary geological, mineralogical and geochemical data about three Ni- and Co-bearing ore deposits located in various parts of Italy and poorly studied so far. We are considering two actual historical mining districts rich in Co and Ni and analogous to "five element vein type" deposits, and comparing them with a peculiar hydrothermal, Cu–Ag mineralization carrying only accessory Co–Ni enrichments but located in a geological context favorable for Co–Ni mineralization at a regional scale. The two historical "five element vein type" mining districts are represented by the Ni-rich Southern Arburese vein system, emplaced in the Palaeozoic basement near the Variscan plutons in the SW part of the Sardinia island, and the Co-rich Usseglio vein system, crosscutting one of the ophiolite complexes in the Western Alpine belt, in Central Piedmont. The third deposit is the poorly known, gabbro-hosted Piazza Cu–Ag-rich orebody located in one of the low-metamorphic ophiolite complexes outcropping in the Northern Appennine belt, Eastern Liguria. The latter deposit exhibits some common features as well as marked differences with the Sardinian and Piedmont counterparts, which had been historically classified as economic for Ni or Co. In spite of this, very little recent bibliographic coverage exists for both the SW Sardinian and the Usseglio veins which can be considered as largely unexplored. As a matter of fact, the Usseglio veins are, at present, object of exploration by international mining companies. In this contribution the three ore systems are, for the first time, evaluated in terms of geological setting, mineral assemblages, geothermometric parameters, and relevant geochemical features, including mineral chemistry and stable isotope signatures of carbonate gangue (where present).

## 2. Materials and Methods

For the Southern Arburese vein system (in SW Sardinia) about 40 mineralized samples were considered from a wider batch deriving from recent mapping and sampling stages in the various old mining sites as well as from historical university and museum collections. The Usseglio vein system (in central Piedmont) was investigated in its structural and mineralogical features by means of detailed mapping and sampling activity resulting in a batch of about 50 samples which were considered for this study. The features of the mineralized stockwork at the Piazza (Ligurian Apennines) were characterized by means of 30 samples deriving from surface as well as underground sampling in the few accessible parts of the old mine. Selected samples underwent mineralogical and micro-textural characterization by optical microscopy under transmitted and reflected light and by scanning electron microscopy (SEM) while quantitative microanalyses of sulfides, sulfosalts, carbonates, and silicates were performed by electron microprobe (EPMA) and scanning electron microscopy with EDS system (SEM-EDS). A further selection provided samples for C and O isotope analyses and microthermometric studies based on fluid inclusions. Details about all the analytical methods and instruments employed are given in Supplementary 1. The tables with the analytical data are for the three mineral deposits also provided as supplementary files.

## 3. Geological, Mineralogical, and Microchemical Features of the Ni–Co Arsenide-Rich Deposits of Sardinia and Piedmont

### 3.1. The Southern Arburèse, Polymetallic Ni–Co–As–Pb–Zn–Cu–Ag–Bi Vein System in SW Sardinia

The Ni–Co-bearing polymetallic veins considered in this section are part of the remarkable, yet poorly studied hydrothermal ore endowment of the Arburèse region in South-Western Sardinia.

The Arburèse ores were first studied during the exploitation of the Montevecchio Pb–Zn–Ag vein district (Figure 1a), one of the most relevant mineral resources of Italy [10,11]. The end of mining activities left unresolved many issues related to the complex metallogenesis of the area [12–14], including the occurrence of superimposed multiple styles of ore mineralization. The southern part of the Arburese region, in particular, is home to a wide variety of deposits including skarns, high-temperature Sn–W–As as well as polymetallic Ni–Co–Pb–Zn–Cu–Ag–Bi–As–Sb veins. The latter, in particular, are uncommon for the Sardinian basement and, like most of the ore deposits in the region, were poorly studied until recent times [15–20].

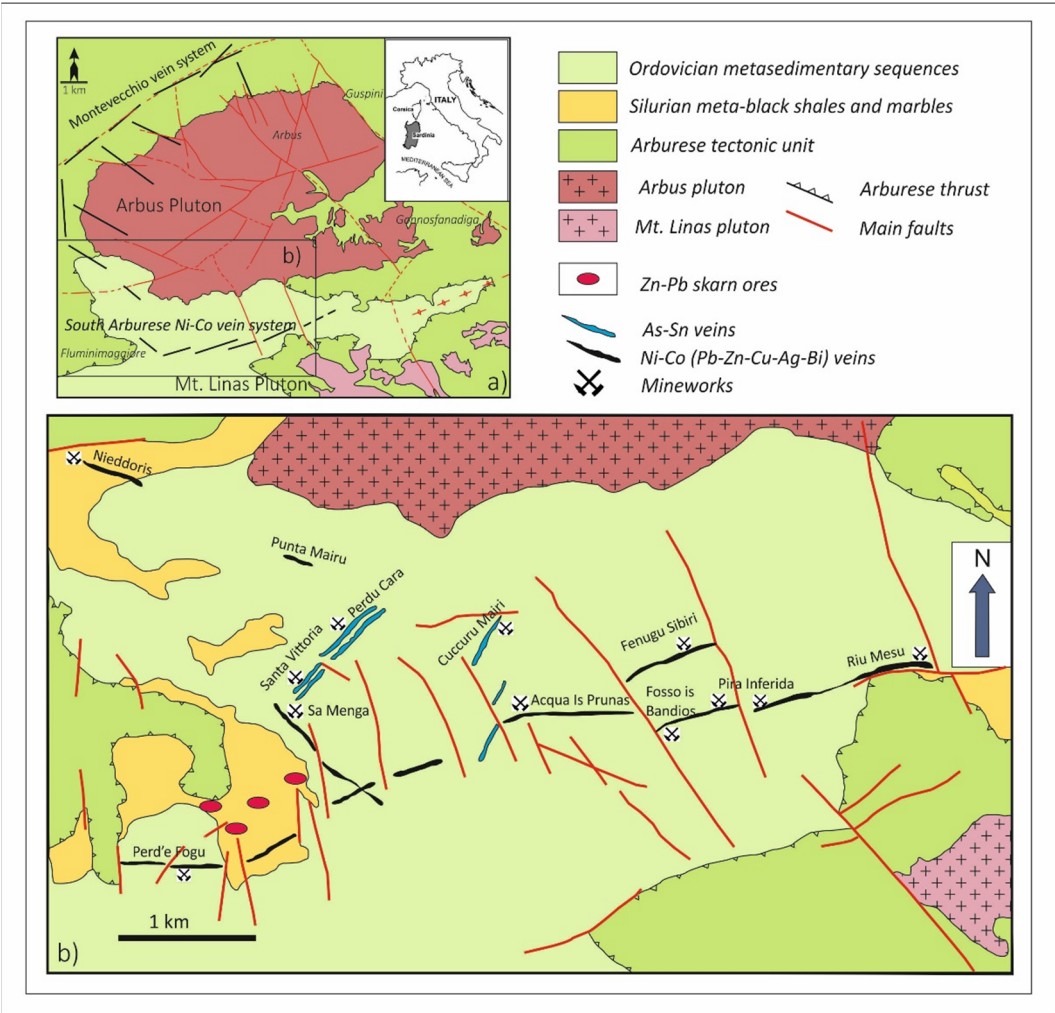

**Figure 1.** Geological setting of the Southern Arburèse vein system in SW Sardinia. (**a**) Schematic geological setting of the Arburèse area with the localization of the Ni–Co-rich vein system (between the Arbus and the Monte Linas plutons) and of the Montevecchio Pb–Zn vein system. (**b**) Simplified geological scheme of the Ni–Co-rich Southern Arburèse vein system with the location of historical mining sites as well as the location of other types of mineral deposits (Zn–Pb skarn and Sn-As veins). After [14,18,19], modified.

These deposits have been sporadically mined for Ni–Co, Pb–Zn–Ag, As, Sn, and W ores during the early 20th century: they have always been considered of minor economic relevance [21], although a modern exploration and evaluation of resources is still lacking. In recent years the interest in this area has greatly grown, since the Arburese region constitutes an obvious exploration target for critical metals. At same time, the region possesses mineral associations that are possible markers for deciphering the tectono-magmatic evolution of Sardinia. The Arburese region displays a complex Variscan-age setting

where sequences structurally framed in the External Zone (or Variscan Foreland) of the Sardinian basement, were overthrust by the Arburese Unit, which is the frontal unit of the Nappe Zone tectonic stack in SW Sardinia [22]. The low-grade metamorphic nappe-foreland complex was intruded by late-Variscan plutons. The polyphase tectonic framework of the region resulted from progressive clockwise migration of shortening directions [23]: (1) initial N–S shortening generated open folds with E–W axial directions; (2) further E–W shortening, contemporaneous to the emplacement of the Arburese tectonic Unit over the Foreland, deformed the previous structures, produced folds with N–S vertical axial planes and penetrative foliation and, in the late stages, backthrusts and asymmetrical folds with variable axial directions.

### 3.1.1. Geology of the Polymetallic Vein System

The polymetallic Ni–Co–Pb–Zn–Cu–Ag–Bi veins, exploited in the old mine areas of Sa Menga, Acqua Is Prunas and Pira Inferida, form an approximately 10-km long ENE–WSW-trending system stretching from the towns of Fluminimaggiore (to the west) to Gonnosfanadiga (to the east) (Figure 1a,b). The vein system is hosted in Paleozoic low-grade metamorphic rocks enclosed between two Variscan granitoid intrusions: the 304 Ma-old Arbus pluton (cordierite-bearing leucogranites, pyroxene-amphibole granodiorites, and quartz-gabbronorites, [14,24]) and the 289 Ma-old Monte Linas Pluton (ferroan, F-bearing monzogranites to leuco-syenogranites; [18,25]). Host rocks to the mineralized veins consist of late Ordovician metapelites and metapsammites of the Portixeddu, Domusnovas and Rio San Marco formations, and Silurian black shales of the Genna Muxerru formation [26] (Figure 1b). From W to E the vein system displays a variable setting, running sideways, at distance of 1–2 km, to the southern margins of the Arbus pluton (Figure 1b), with NW–SE to ENE–WSW direction. Overall, the system is formed by an alignment of several 1–2 m thick, high-angle veins dipping almost constantly southward: this structural arrangement and the "peri-plutonic" trend are among the many similarities with the main Montevecchio vein field to the north [17,20]. Along the development of the vein system, the single orebodies are quite discontinuous, essentially consisting of numerous oreshoots alternating with poorly mineralized zones. The main outcrops are exposed on a series of small mines that followed the main direction and southern dipping of the system, i.e., from west to east, the Nieddoris, Sa Menga, Acqua Is Prunas, Pira Inferida, and Riu Mesu mines (Figure 1b). Located at the South-Western end, the 4 km-long, EW-trending and northwards dipping Pb–Ag-rich Perd'E'Fogu quartz vein was considered as a lateral branch of the system. In most of the mining sites, mineralized veins are in prevalence hosted by the Ordovician metasediments, with the exception of the Acqua Is Prunas mine area, where the km-long veins are enclosed in Silurian black shales. As mentioned above, the metallogenic picture of the area also includes several high-temperature cassiterite- and arsenopyrite-rich vein sets, the most important of which were exploited at the Perdu Cara [18] and Santa Vittoria mine sites (Figure 1b). These veins (presently still under investigation) display intersection relationships with the polymetallic vein system in some of the main Ni–Co locations, e.g., near Sa Menga and Acqua is Prunas mines.

### 3.1.2. Ore Assemblages, Textures, and Micro-Chemical Features

The Ni–Co-bearing polymetallic veins, still accessible inside some of the old workings and in outcrops, exhibit open space filling, brecciated and, occasionally, banded textures (see Figure 2a–f). Ore minerals occur as Ni–Co-rich nodules, cockades and stockworks associated to Fe carbonates and subordinate microcrystalline quartz (Figure 2b,c) or in quartz (Figure 2d), and as Ni–Co-poor, base metal sulfide-rich aggregates, bands and disseminations in late carbonate and comb quartz veinlets (Figure 2e,f).

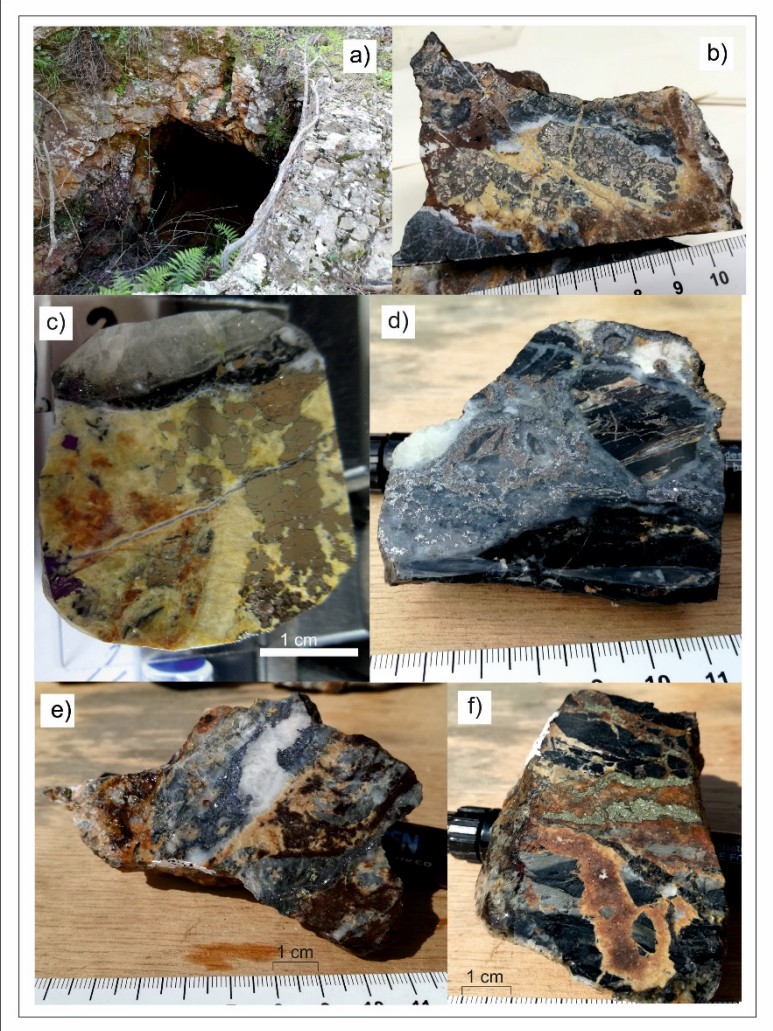

**Figure 2.** Outcrop and macroscopic textures of the mineralization from the SW Sardinian veins. Montevecchio mining district. (**a**) Entrance of an old adit at the Sa Menga mining site with exposed carbonate-rich mineralization (rusty patches); (**b**) cockade and brecciated ore with Ni–Co-rich aggregates, showing visible niccolite cores, overgrown by siderite and quartz and crosscut by later carbonate veinlets (Sa Menga); (**c**) Ni–Co-rich cockade nodules in siderite and crosscut by quartz veinlets, and with fragments of Silurian black shale (Acqua Is Prunas); (**d**) brecciated ore with large fragments of black shale and arborescent aggregate of Ni–Co arsenides in quartz (Pira Inferida); (**e**) galena-rich (with sphalerite and tetrahedrite) ore in quartz veinlets cementing brecciated siderite-rich patches (Sa Menga); (**f**) chalcopyrite- and, to the right, sphalerite-rich bands with quartz crosscutting brecciated black shale cemented by siderite (Sa Menga).

Brecciated textures involving wallrock fragments are frequent as well, e.g., towards veins selvedges or in zones of interaction of different veins. Typical ore cockades at Sa Menga and Acqua Is Prunas (Figure 2b,c) consist of mm- to cm-sized spheroidal nodules made of Ni–Co sulfosalts and encrusted by siderite and minor microcrystalline quartz. At Pira Inferida zoned cockade aggregates may be accreted by microcrystalline quartz (Figure 2d). Also, the base metal sulfide aggregates may locally display some mineral zoning (see below). Common wallrock alteration includes zones of sericitization and silicification. Previous studies outlined the complexity of the mineral assemblages including Ni–Co arsenides and sulfarsenides, base metal sulfides, sulfosalts and native elements. Within the main vein system these assemblages show a broad continuity, even though in different mine localities ores may display significant variations in the relative amount of mineral species. Accordingly, in the westernmost Nieddoris mine Ni (Co) sulfarsenides were reported to prevail over Ni–Co arsenides and

over sulfides, and in Sa Menga and Riu Mesu mine base metal sulfides were relatively more abundant than Ni–Co arsenides and sulfarsenides. Conversely, Ni–Co arsenides and sulfarsenides are listed as dominant in Acqua is Prunas mine. To the east, for the Pira Inferida mine old mine reports recorded grades of 14 wt% Ni and 6 wt % Co in the arsenide–sulfarsenide ore [27]; the same Ni (14%), but much lower Co grades (0.2%) can be found in very old reports from the Perd'e Fogu vein (Perda S'Oliu mine) in the western end of the mining district [28].

The samples selected for this study contributed to outline the paragenetic scheme of the mineralization across the vein system (Figure 3a) and confirmed the complexity of mineral assemblages. In the different veins mineralization includes three stages of formation: (a) an early stage with Bi (native Bi and bismuthinite, Bi$_2$S$_3$); (b) an As–(Sb)–rich stage with various Ni–Co arsenides/antimonides (nickelite–breithauptite, rammelsbergite, safflorite, and skutterudite); and sulfarsenides (gersdorffite–cobaltite with accessory ullmannite) accompanied with native Au; (c) a S-rich stage with abundant base metal sulfides (sphalerite, galena, chalcopyrite, and rare pyrite), Ag-bearing tetrahedrite, bournonite and accessory, late-stage acanthite, and Ni–Co-bearing sulfides (vaesite, millerite).

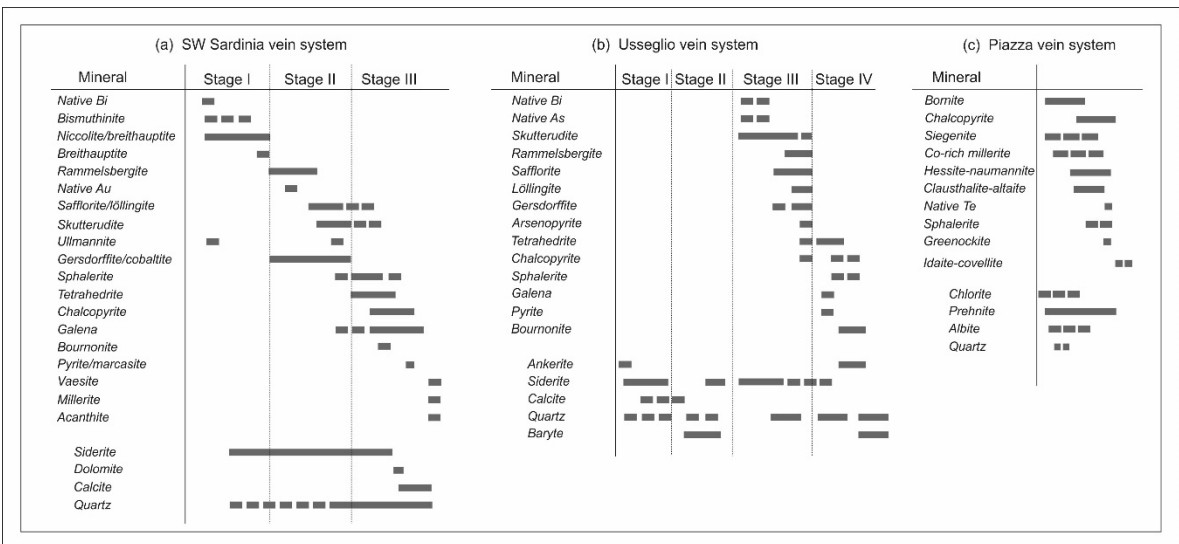

**Figure 3.** Paragenetic schemes for the mineral deposits considered: SW Sardinia (**a**), Usseglio (**b**), Usseglio and Piazza (**c**).

Gangue in stage (a) and (b) is represented by siderite and minor microcrystalline and fibrous quartz, while stage (c) is still marked by the presence of siderite with minor calcite and rare dolomite, followed by comb-textured quartz. Siderite usually is in large aggregates of idiomorphic crystals, often showing graphic textures with quartz (Figure 4a), while rare dolomite is skeletal and overgrown by late calcite (Figure 4b). Sulfides are frequently associated with quartz-rich stockwork (Figure 4c). It is worth mentioning the occurrence of disseminations of fine-grained REE carbonate synchysite Ca(Ce,La)(CO$_3$)$_2$F reported by [29] from quartz gangue. Ore microscopy, scanning electron microscopy (SEM) imaging and microchemical analyses documented for the arsenide stage a prevalence of Ni-rich phases, with the initial crystallization of mono-arsenides and accessory mono-antimonides, followed by di-arsenides, sulfarsenides, and tri-arsenides (Figure 4d–o; Supplementary Table S1). Thus, in the typical zoned arsenide nodules of this mineralization, the mono-arsenide/mono-antimonide solid solution nickelite-breithauptite Ni (As,Sb) builds up the cores characterized by zoned, radiated to fibrous Sb-bearing niccolite crystals and aggregates overgrown by sulfarsenides and diarsenides (Figure 4d,g,i and Figure 5a,c,f).

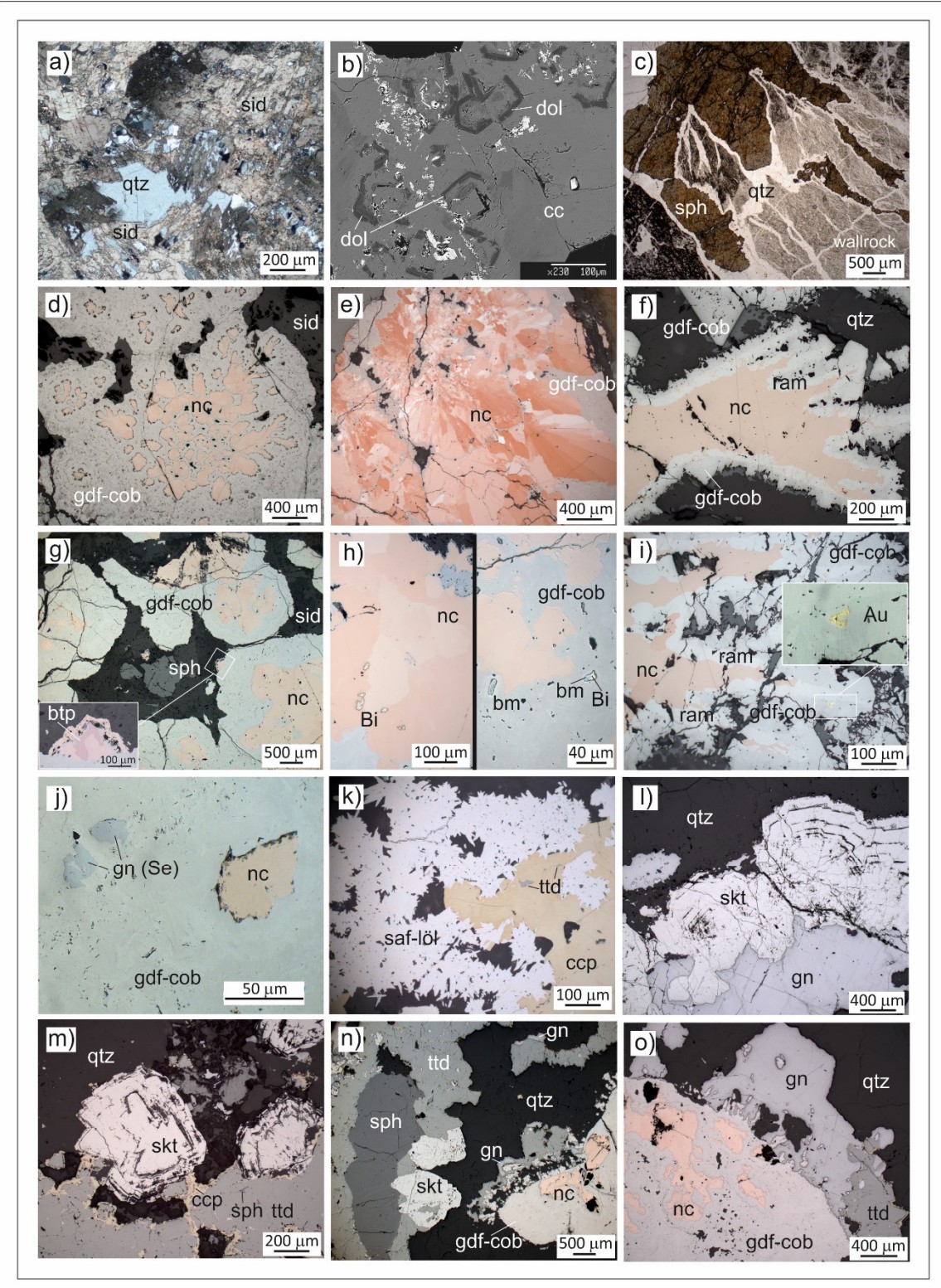

**Figure 4.** Microscopic features of the mineralization from the SW Sardinian veins. (**a**) Symplectite-like intergrowths of quartz and euhredral siderite associated with Ni–Co arsenide nodules—Pira Inferida. (**b**) Late-stage calcite cement accompanying sphalerite mineralization and enclosing skeletal dolomite—Sa Menga. (**c**) Brecciated, silicified shaly wallrock cemented by quartz and sphalerite, brown-Sa Menga. (**d**) Ni–Co-rich nodules with niccolite cores overgrown by Ni–Co sulfarsenide-rich cockades—Acqua Is Prunas. (**e**) Radial to plumose structure of the niccolite cores—Sa Menga (crossed

nicols). (**f**) Ni–Co-rich aggregates with wide niccolite interior and rims of rammelsbergite (inner) and Ni–Co sulfarsenides (outer)—Pira Inferida. (**g**) Ni–Co-rich nodules with niccolite cores and thick Ni–Co sulfarsenide-rich cockades, with external breithauptite aggregate and sphalerite in carbonate gangue—Sa Menga. (**h**) Inclusions of native Bi and bismuthite in niccolite cores as well as in sulfarsenide-rich crusts—Sa Menga. (**i**) Native gold in outer Ni–Co sulfarsenide crusts—Pira Inferida; (**j**) inclusions of seleniferous galena in Ni–Co sulfarsenide-rich cockades—Acqua Is Prunas. (**k**) Aggregates of star-like safflorite and löllingite with chalcopyrite including small specks of tetrahedrite—Pira Inferida. (**l**) Zoned, fan-shaped skutterudite aggregates partly replaced by galena—Sa Menga. (**m**) Concentrically zoned skutterudite and tetrahedrite rimmed with chalcopyrite—Sa Menga. (**n**) Fractured niccolite–sulfarsenide nodules and zoned skutterudite aggregates overgrown by base metal sulfides showing a depositional sequence from sphalerite to tetrahedrite to galena—Pira Inferida. (**o**) Fractured niccolite–sulfarsenide nodule deeply corroded by galena–tetrahedrite—Sa Menga. (a,c): plane polarized transmitted light; d, e, f, g, h, i, j, k, l, m, n, o: plane polarized reflected light; b: electron microscopy backscattered images). Abbreviations: bm = bismuthite, btp = breithauptite, cc = calcite, cob = cobaltite, ccp = chalcopyrite, dol = dolomite, gdf = gersdorffite, gn = galena, löl = löllingite, nc = niccolite, qtz = quartz, ram = rammelsbergite, saf = safflorite, sid = siderite, skt = skutterudite, sph = sphalerite, ttd = tetrahedrite, ul = ullmannite.

Small breithauptite aggregates may also grow over the outer sulfarsenide rims (Figure 4g). At Sa Menga and Acqua is Prunas the thick niccolite cores are overgrown by rhythmic crust-like microlayers of gersdorffite–cobaltite (Ni,Co)AsS solid solution, locally probably intermixed with fine-grained rammelsbergite–safflorite (giving rise to "mixed" EPMA analyses). The concretions may terminate with a late growth of ullmannite (NiSbS) and vaesite, $NiS_2$ of probable secondary origin (Figure 5c,f). At Pira Inferida the arborescent niccolite aggregates are overgrown by an inner rim of safflorite–rammelsbergite $(Co,Ni)As_2$ and outer gersdorffite–cobaltite aggregates (Figures 4f–i and 5b). The niccolite cores as well as the diarsenide–sulfarsenide crusts often include fine-grained early-stage native Bi and/or bismuthinite blebs (Figure 4h) and rare ullmannite grains of early deposition. Small gold grains were observed in the diarsenide–sulfarsenide rims over the niccolite cores at Pira Inferida (Figure 4i). In some cases (Acqua is Prunas and Sa Menga) the sulfarsenide rims host small inclusions of Se-rich (up to 4 wt%) galena (Figure 4j; Table S1). Star-like aggregates of safflorite and löllingite (occasionally overgrown by arsenopyrite) also occur in the Sa Menga and Pira Inferida ores (Figure 4k). In these ores Ni and Co are also concentrated in fan-shaped aggregates of triarsenide skutterudite (Figure 4l–n), with composition between Ni–skutterudite $(Ni,Co)As_{3-x}$, and Fe–skutterudite $(Fe,Co)As_3$ (Table S1, Figure 6a). Skutterudite nodules, safflorite aggregates as well as rammelsbergite–sulfarsenide cockades are variably overgrown and corroded by base metal sulfides and tetrahedrite (Figure 4k–o). Moderate brecciation of the previous assemblages, cemented by quartz ± carbonate gangue (Figure 4c), generally marks the advent of the sulfide stage. Sphalerite, followed by abundant tetrahedrite (Cu, Fe, $Ag)_{12}Sb_4S_{13}$), galena, chalcopyrite, minor bournonite ($PbCuSbS_3$) and accessory acanthite ($Ag_2S$) characterize this stage; pyrite is rare, while accessory millerite may occur in late veinlets.

Sphalerite also occurs as accessory grains interstitial to the niccolite-cored nodules (Figure 4g) in the Ni–Co-rich, sulfide-poor ore facies and is particularly abundant in the Sa Menga veins. ISphalerite is transparent and shows a slight but distinct anomalous anisotropy. Galena frequently displays diffuse, often very abundant (sub-)micro-inclusions (Figure 5d,e). These features of sphalerite and galena are in common with the Montevecchio ores [30]. Some of the galena-hosted microinclusions, amenable for analysis, consist of tetrahedrite and bournonite. Tetrahedrite is low in As and displays variable composition (Table S1). The occurrence of Ag-rich compositions close to freibergite and accessory acanthite blebs indicate that the sulfide stage was also marked by a sensible Ag enrichment, in agreement with old data by [16], who also reported the presence of stephanite ($Ag_5SbS_4$) and proustite–pyrargirite ($Ag_3AsS_3$–$Ag_3SbS_3$) in the ores.

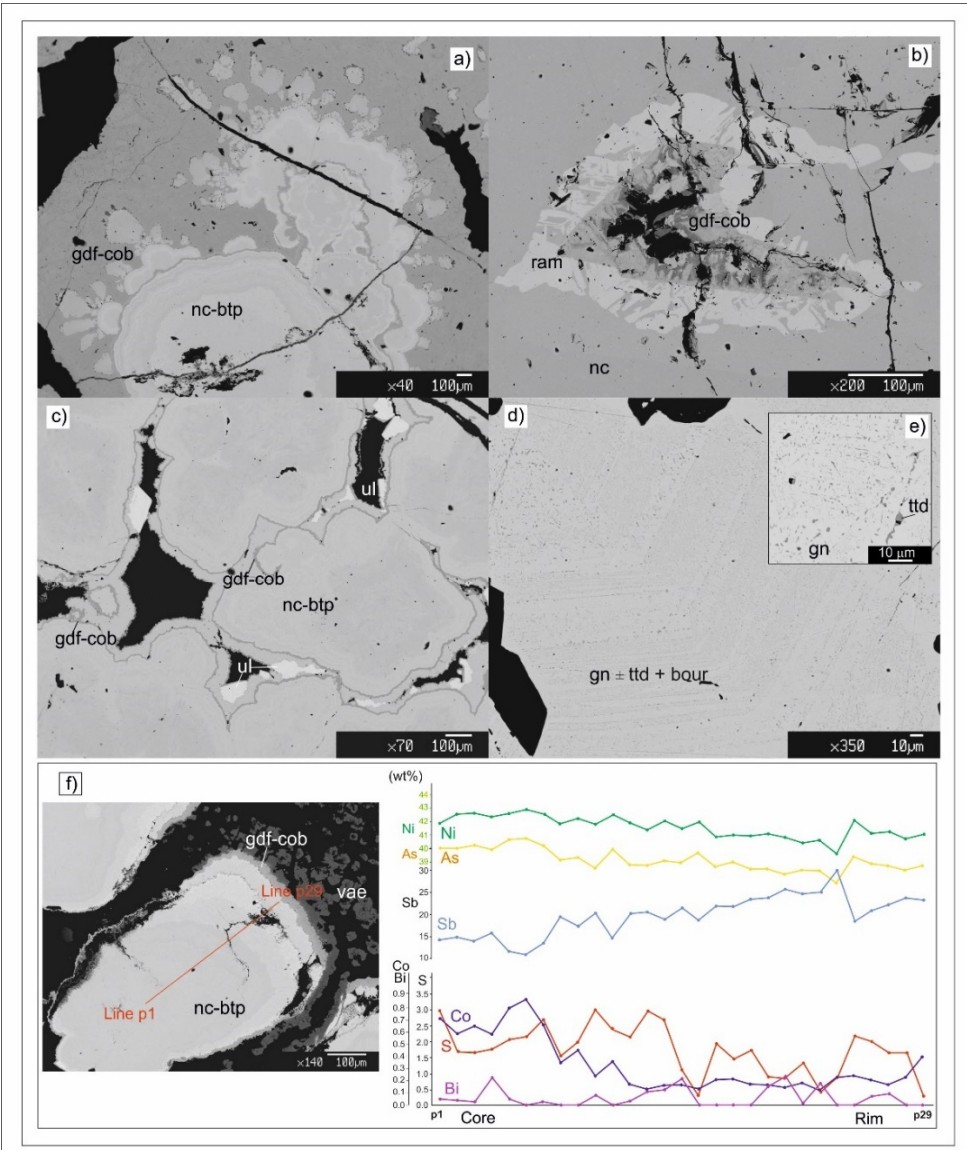

**Figure 5.** Selection of electron microscopy backscattered images from the SW Sardinian veins. (**a**) concretionary and zoned texture of niccolite nodules and their outer crusts made mostly of thin layers of gersdorffite–cobaltite—Acqua Is Prunas; (**b**) portion the arborescent niccolite concretions of Figure 4f with a cavity lined by rammelsbergite and then by gersdorffite–cobaltite—Pira Inferida; (**c**) zoned, Sb-bearing niccolite nodules with thin crusts of gersdorffite–cobaltite and with late growth of euhedral ullmannite—Acqua Is Prunas; (**d**) galena highly enriched in (sub-)micrometric inclusions of sulfosalts according to more or less regular patterns—Pira Inferida; (**e**) inset of (**d**) showing a portion of galena with relatively coarser sulfosalt inclusions amenable for EPMA analysis (tetrahedrite)—Pira Inferida; (**f**) SEM image of a niccolite–breithauptite nodule showing a progressive zoning (shades of grey-white) from core to rim, with an outer crust of gersdoffite–cobaltite and growth of secondary vaesite (vae) crystals (Acqua Is Prunas). The red line indicates the position of 29 data points for the analytical profiles for Ni, As, Sb, Co, and Bi shown in the nearby diagram. The Bi peaks may represent submicrometric native bismuth inclusions. Abbreviations: bour = bournonite, btp = breithauptite, cob = cobaltite, gdf = gersdorffite, gn = galena, nc = niccolite, ram = rammelsbergite, ttd = tetrahedrite, ul = ullmannite, vae = vaesite.

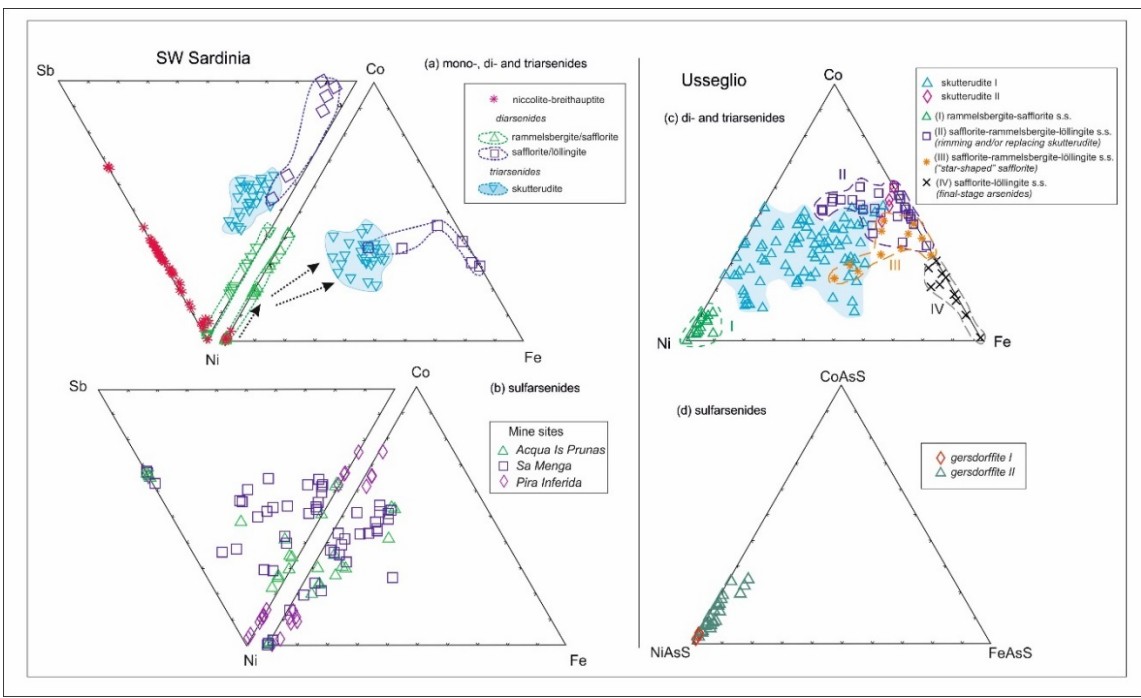

**Figure 6.** Plots of the data of arsenides and sulfarsenides (atomic%). (**a**,**b**) ternary diagrams comparing Ni, Co, Fe, and Sb components for mono-, di-, and triarsenides and for sulfarsenides–sulfantimonides in the SW Sardinian vein system; (**c**,**d**) ternary diagrams comparing Ni, Co, and Fe components for di- and triarsenides and for sulfarsenides in the Usseglio vein system.

EPMA analyses on the Co–Ni phases in the SW Sardinian veins are summarized in Figure 6a,b. The diagram in Figure 6a gathers the di- and triarsenides and illustrates the progressive enrichment in Co and Fe recorded during the transition from mono- to diarsenides and triarsenides as well as the characteristic dual enrichment in Sb both in the niccolite nodules and, to a lesser extent, the skutterudite aggregates. Some niccolite data plotted are from a core-rim transect across a zoned nickeline nodule (from Acqua Is Prunas; Figure 5f): in particular, the profiles document a tendency to inner Co-richer character and outwards Sb enrichment, compatible with the occurrence of marginal breithauptite aggregates and ullmannite crystals. The diagrams in Figure 6b display the chemical variability of the sulfarsenides and sulfantimonides occurring in the three main mining areas: the phases are homogeneously distributed between Ni and Co endmembers, but the terms closer to cobaltite (recorded at Pira Inferida) are least enriched in Sb. The sphalerite composition is shown in Figure 7a–d (Table S1). Sphalerite contributes to distinguish the Ni–Co–As-bearing mineralization from the high-temperature arsenopyrite-rich ore of the Santa Vittoria vein system (Figure 1b). The latter contain accessory sphalerite which is plotted for comparison. The Santa Vittoria high-temperature vein sphalerite is much higher in Fe than the Ni–Co-related sphalerites (Figure 7a). These display variable yet locally notable Cd contents (Figure 7b) which may justify the anomalous anisotropy locally observed and interpreted by Seal (1985) as resulting from hexagonal, wurtzite sub-domains stabilized by Cd and Fe in the cubic sphalerite structure. Sphalerite records Ag contents above detection (up to 0.18 wt%) probably related to sub-micro-inclusions of Ag phases. Also, frequent, random Ge enrichments (up to 0.19 wt%) are observed, occasionally paired with irregular Co peaks (Figure 7c). Selenium above detection level was locally recorded in some sphalerite grains (max 0.07 wt%; Table S1) (Figure 7d).

The Se and other accessory element contents and distribution in sphalerite need to be further investigated. EPMA on carbonates have been performed on siderite, calcite and rare dolomite in samples from Sa Menga and Acqua Is Prunas (Table S1). Among them, siderite stands out for sensible chemical variability involving Mg, Ca, and Mn ($Sd_{78.4–89.9}Mgs_{4.7–15.6}Rds_{1.5–8.9}Cal_{0.3–6.3}$) and recording locally

detectable Zn contents (max 0.2 wt% ZnO). Calcite is impure ($Cal_{90-99.3}Mgs_{0.1-3.3}Sd_{0.6-3.24}Rds_{0.18-1.04}$) with irregular, mild Zn enrichments (max 2.9 wt% ZnO).

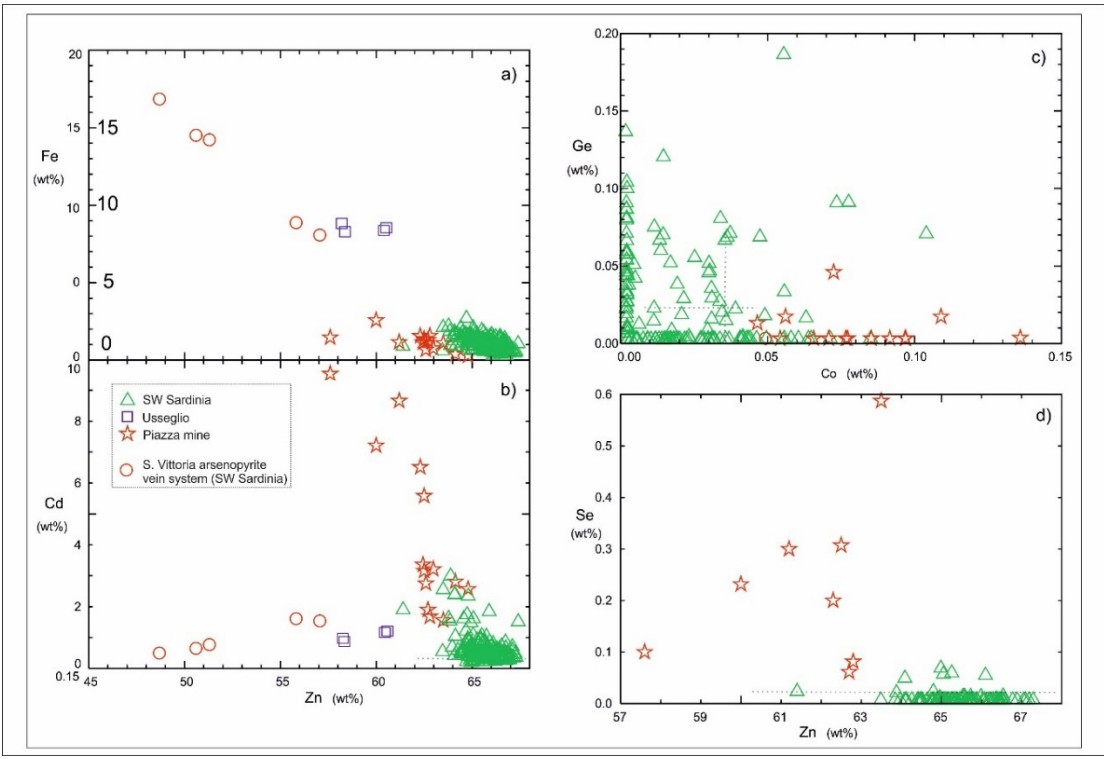

**Figure 7.** Plots of the sphalerite data (weight%). (**a,b**) binary diagrams of Zn vs. Fe and Cd involving sphalerites from SW Sardinian, Usseglio and Piazza deposits and comparing them with the composition of accessory sphalerite from arsenopyrite veins affiliated to the high-temperature Sn–Mo–As ores near Sa Menga [18]. (**c,d**) binary diagrams comparing Ge, Co, and Se contents in sphalerites from the SW Sardinian and Piazza ore deposits.

### 3.2. The Usseglio Co–Ni-Rich Vein System in Piedmont, Western Alps

The Italian Western Alps are host to a number of post-orogenic hydrothermal deposits which were exploited in the past. The best known are the Brosso and Traversella mines related to the Tertary magmatism and the orogenic mesothermal gold-bearing quartz lode systems (including the historical Monte Rosa Gold District; [31] with its references). Little is known, instead, about polymetallic vein systems characterized by the abundance of Fe-rich carbonate associated with a variable metal content including Co, Ni, Cu, Pb, Au, Ag, As, and Sb. In particular, siderite-rich hydrothermal veins characterized by the occurrence of Co–Ni–Fe arsenides mineralization occur in the Viù valley (the southernmost of the three Lanzo valleys, NW Piedmont), in the Usseglio municipality. For many decades these ore deposits were almost completely forgotten in the geological literature, after some studies performed during the first half of the last century [32,33]. Recently new studies have been carried out both on the geological and metallogenic features of these deposits and on the historical and archeological aspects connected with mining in the Usseglio area [34,35]. These studies have documented that the veins system, also named Punta Corna mining complex, was mined for iron, and locally for silver, already in the middle ages (13th–15th centuries). The exploitation of the cobalt ore began, instead, in the mid-18th century. In this paper a review of the recent studies on the hydrothermal system is presented, as well as unpublished data on the cobalt ore composition.

### 3.2.1. Geological Setting and Field Relationships of the Usseglio Veins

The Usseglio hydrothermal system crops out on the left side of the Viù Valley (in the Arnàs lateral valley) and subordinately, to the north-east, in the adjacent Ala Valley. The system is hosted by the Piemonte Zone which is composed, in this sector of the Lanzo Valleys, of metaophiolitic rocks (metabasalts with minor metagabbros, ultramafic rocks and minor metasediments) representing disrupted sections of oceanic lithosphere of Jurassic age [36–38] (Figure 8a). Such sequence records a multistage tectono–metamorphic evolution: after an early "oceanic" stage, the alpine evolution started with the early-Alpine metamorphic stage, under eclogitic conditions, followed by a decompressional P-T-t path, with a final re-equilibration under greenschist–facies conditions [39]. The subsequent, composite structural evolution developed under brittle conditions [40,41]. The mineralized Usseglio veins are hosted by E–W-trending, subvertical brittle post-metamorphic structures. Host rocks to the hydrothermal veins are greenschist–facies metabasites and, in one locality only, strongly foliated calcschists. All along the hydrothermal system the veins sharply crosscut the foliation of the wallrock, which is always affected by intense sericite–quartz–carbonate hydrothermal alteration [42].

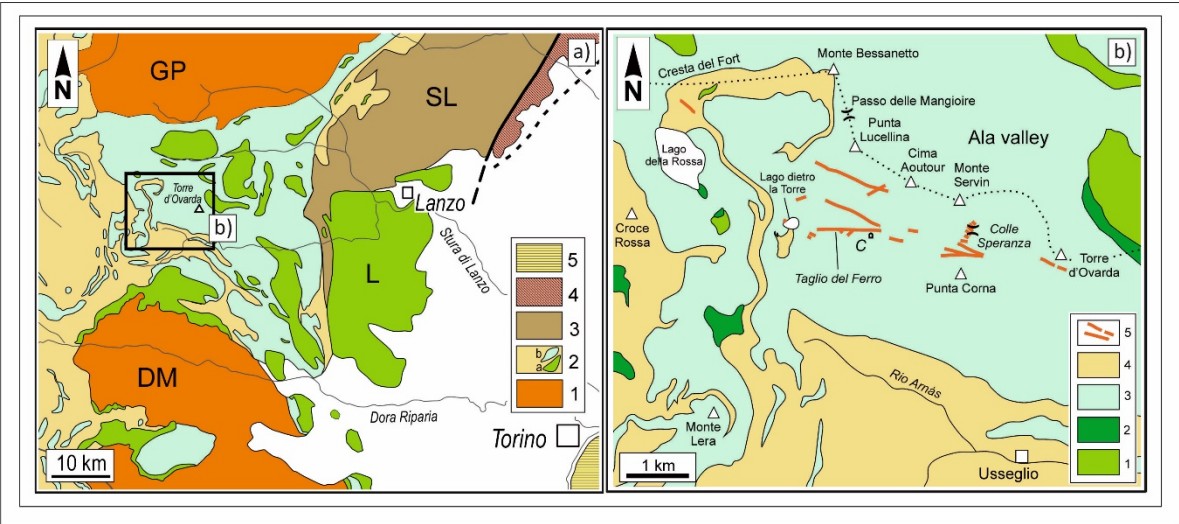

**Figure 8.** (**a**) Simplified tectonic sketch-map of the central sector of the Western Alps. 1: internal crystalline Massifs of Gran Paradiso (GP) and Dora Maira (DM); 2: Piemonte Zone (a: main ultramafic bodies, b: metabasites, L: Lanzo ultramafic massif); 3: Austroalpine Domain (SL: Sesia Lanzo Zone), 4: Southalpine Domain. 5: late- and post-orogenic deposits of the Torino hill. (**b**) Simplified geological map of the Arnàs valley (Usseglio). 1: serpentinites, 2: metagabbros, 3: metabasites, 4: metasediments (calcschists, quartzites, marbles). 5: main hydrothermal veins. C: entrance of the crosscut at 2374 m. After [41], modified.

The mineralized vein system is developed over a total length of 6–7 km along WNW–ESE direction and a width of ca. 1 km, at an altitude ranging from 2250 and 2900 m (Figure 8b). Major single veins show a total length of about 1 km and a maximum thickness up to 6–7 m. Within the fracture field vein trends can vary from E–W to WNW–ESE (and locally WSW–ENE), with steep dipping ca. 80° to the north to 60–70° to the south in the western and eastern sector, respectively. The main veins have a strong morphological evidence in the landscape, due to trenches related to historical surface mining excavations for iron ("Taglio del Ferro", i.e., "Iron cut" in Figure 9a). However, mineralized outcrops are rare: in fact, the thickest veins are marked by trenches, whose bottom is covered with debris, dump materials and vegetation. Therefore, the mineralization features can be observed mainly from loose blocks and along few accessible mine galleries, e.g., the crosscut at 2374 m (C in Figure 8b). Such mine galleries reveal that the mineralization at least locally consisted of dense swarms of cm-to dm-thick subparallel veinlets crosscutting the strongly altered metabasite host (Figure 9b). The Co–Ni arsenide mineralization is not evenly distributed within the vein system, as it characterizes especially the Punta

Corna veins in the eastern sector. The western sector is instead mainly composed of siderite and Co–Ni minerals are virtually lacking. At surface the transition between siderite and the cobalt mineralization styles is not exposed. However, the spatial arrangement of the veins (Figure 8b) suggests that the Punta Corna veins are the prosecution of the western siderite-rich vein system. The hydrothermal system appears to further continue both eastwards and westwards with the poorly known veins in the sites of Torre d'Ovarda, Punta Lucellina and Lago della Rossa (Figure 8b). In particular, a steep, ESE–WNW-trending, massive siderite–quartz vein close to Lago della Rossa, which is the only one hosted by calcschists, marks the western termination of the outcropping system (Figure 8b).

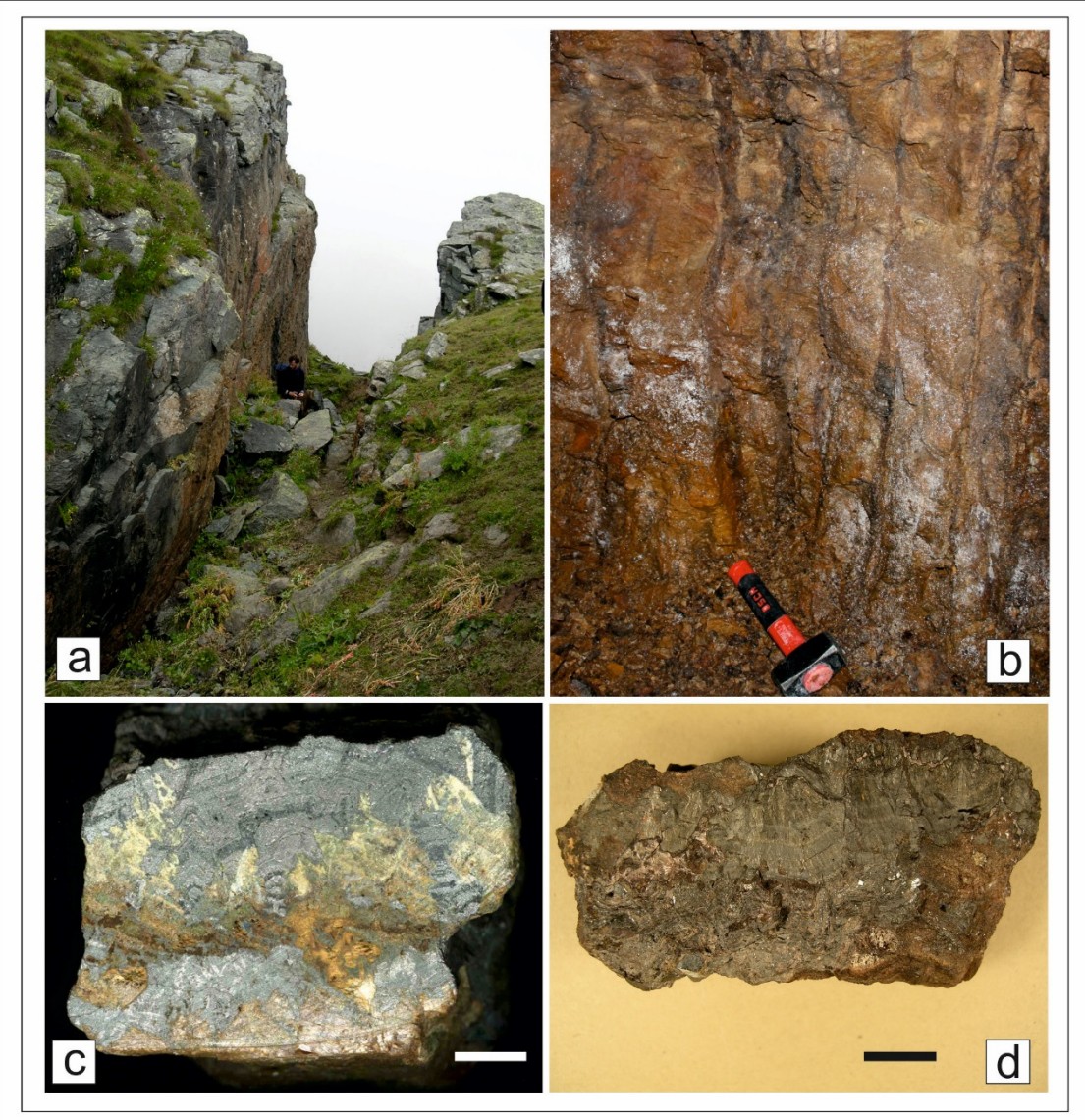

**Figure 9.** Usseglio vein system: (**a**) morphological depression due to the mining excavation for iron along the main siderite-rich vein ("Taglio del Ferro"); (**b**) underground exposure of a main siderite-rich vein along the crosscut at 2374 m. The "vein" actually consists of a series of subparallel, centimeter- to decimeter-thick veins crosscutting strongly altered metabasites. (**c**,**d**): hand samples of the Co–Ni–Fe arsenides mineralization, (**c**): coarse-grained skutterudite (metallic grey) showing crystal growth structures, scale bar: 1 cm; (**d**): coarse-grained, fan-shaped rammelsbergite, scale bar: 2 cm.

### 3.2.2. Mineral Assemblages and Chemical Composition of Minerals

At Usseglio mineralization displays variable characters and textures in the different sectors of the vein system. In the western sector, mineralization typically displays massive texture and is largely composed of Fe carbonates with minor quartz and calcite and scarce sulfides, except at the Lago della Rossa vein, where the massive Fe carbonate-rich vein filling is affected by fine-grained dissemination of sulfides (pyrite, chalcopyrite, tetrahedrite) and hematite. Massive siderite is locally brecciated and cemented by abundant baryte ± quartz. Eastwards, a transition occurs from massive to banded texture in the veins associated with the occurrence of the Co-rich mineralization (Figure 9c,d). The major banded veins show thin quartz selvedges and Fe carbonates (± minor quartz) filling, or a siderite zone passing to comb quartz ± baryte-rich cores. Baryte-rich cores may display coarse-grained patches composed of cm-sized baryte crystals associated with minor quartz. Veins locally display brecciated textures, with up to cm-sized fragments of strongly altered metabasite or vein material (Fe carbonates, baryte, and/or quartz) cemented by quartz and siderite. Metabasite clasts encrusted by quartz and siderite give way to cockade textures. In the Punta Corna area, coarse-grained Co–Ni–Fe arsenide mineralization (Figure 9c,d) may occur both in the inner part of banded veins, associated with Fe carbonates and quartz, and as massive veinlets crosscutting the altered metabasite. Skutterudite locally occurs as centimetric euhedral crystals; rammelsbergite may form fan-shaped intergrowths (Figure 9c). Arsenides occur as complex intergrowths, with skutterudite associated with and partly replaced by di-arsenides. Accessory base metal sulfides and tetrahedrite may also occur, and in some cases clearly postdate the arsenides.

The combination of field data with ore petrography and mineral chemistry allows the recognition of four main deposition stages (older to younger): (1) siderite, (2) baryte, (3) Co–Ni–Fe arsenides, and (4) sulfides stage (Figure 3b).

(1) The siderite-dominated stage begins with the deposition of ankerite (ankerite *I*, western sector) or comb quartz (mainly eastern sector), as a millimeter-thick rim at the contact with the host metabasite, followed everywhere by abundant medium-grained siderite (siderite I), with minor interstitial quartz and calcite. The thickest veins of the western sector are almost completely composed of siderite and quartz belonging to this stage.

(2) The baryte stage is characterized by the precipitation of baryte and minor quartz. Baryte is found at core of siderite-rich banded veins or within breccia bodies, where it constitutes the matrix of siderite clasts. Baryte postdates siderite I, but at least locally shows equilibrium relationships with a Fe-carbonate (siderite II), which typically displays a bladed shape.

(3) The Co–Ni–Fe arsenides stage is characterized by the deposition of abundant Co-rich, Ni–Fe-bearing tri- and di-arsenides, and minor native As and Bi ± accessory base metal sulfides and tetrahedrite. The beginning of this stage is generally marked by the deposition of skutterudite I (Figure 10a), as euhedral crystals often showing well developed growth zoning associated with medium-grained siderite (siderite III) and minor interstitial quartz. Native arsenic and bismuth may occur at core of skutterudite. Native arsenic occurs as intergrowths of tabular to dendritic crystals, often enclosing few microns-sized rounded crystals of native bismuth; the latter also forms globular-shaped intergrowths, suggesting coalescence of liquid droplets. Gersdorffite I locally occurs along growth zones in skutterudite. The latter often shows an outer primary rim composed of di-arsenides (safflorite–rammelsbergite–löllingite s.s.). More often, it is replaced by di-arsenides and/or sulfarsenide (gersdorffite II). Within cm- to dm-sized domains almost completely composed of arsenides, skutterudite is replaced by gersdorffite which is, in turn, overgrown by star-shaped twins of safflorite-löllingite s.s. (Figure 10b,d,e). Skutterudite occurring as isolated euhedral crystals in siderite III is, instead, more often partially overgrown by rammelsbergite (Figure 10c). Very rarely, skutterudite II occurs as thin veinlets which crosscut the diarsenides. Late-stage di-arsenides are locally intergrown with minor amounts of tetrahedrite I and sulfides (mostly chalcopyrite with accessory sphalerite).

(4)     The final hydrothermal stage is characterized by the abundance of sulfides and sulfosalts, represented by tetrahedrite II, pyrite, chalcopyrite, scanty sphalerite, galena, and bournonite (Figure 10f) associated with variable amounts of quartz and Fe carbonate (siderite IV). These phases locally occur as coatings of the arsenides, or breccia cement of the earlier assemblages. Tetrahedrite II is locally abundant and is partially replaced by bournonite. Locally, late-stage scattered ankerite II or quartz + baryte veinlets occur, which cement clasts of the earlier assemblages.

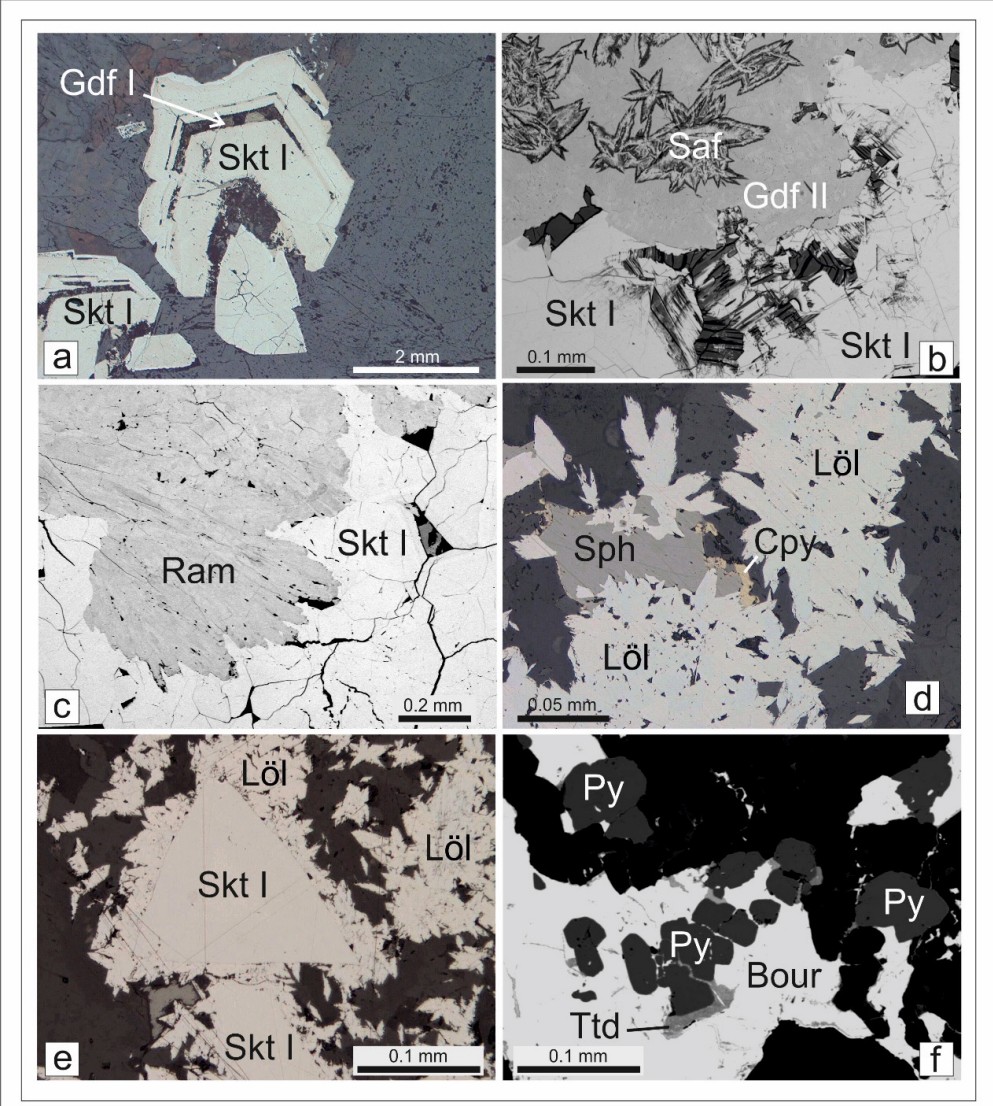

**Figure 10.** Microstructural features of the Co–Ni–Fe arsenides and sulfides mineralization. (**a**): coarse-grained skutterudite showing well developed growth zoning. Gersdorffite I occurs along some growth zones; (**b**): skutterudite I is partially replaced by gersdorffite II which is, in turn, overgrown by star-shaped "safflorite twins"; (**c**): skutterudite I overgrown by rammelsbergite; (**d**) late stage löllingite associated with sphalerite and chalcopyrite; (**e**) skutterudite I embedded by löllingite; (**f**) final stage bournonite embedding tetrahedrite and pyrite. (a, d, e, f: plane polarized reflected light, b, c: electron microscopy backscattered images). Abbreviations: Ccp = chalcopyrite, Gdf = gersdorffite, Löl = löllingite, Py = pyrite, Ram = rammelsbergite, Saf = safflorite, Skt I = skutterudite, Sph = sphalerite, Ttd = tetrahedrite.

The Fe carbonate-rich vein filling displays distinct composition in the different stages (Table S2). Siderite I is a siderite–magnesite s.s., with very minor rhodochrosite and calcite components

($Sd_{66.4-73.1}Mgs_{23.8-29.1}Rds_{2.3-2.9}Cal_{0-2.6}$). Siderite III of the Co–Ni–Fe arsenide stage displays a wide composition range ($Sd_{70.7-90.8}Mgs_{4.4-26.5}Rds_{2.4-4.9}Cal_{0-3.4}$) and is mostly enriched in Fe and depleted in Mn and Ca compared with siderite I.

Arsenides show wide variations in composition (Table S2). Skutterudite I composition mostly plots within the solid-solution field defined by [43] and is characterized by a moderate to low Co content ($CoAs_3 \leq 53\%$) and a wide range of Ni/Fe ratios ($Co_{0.11-0.53}Ni_{0.07-0.75}Fe_{0.03-0.51}$) (Figure 6c), much wider than for the minor Sardinian skutterudite. Coarse-grained zoned crystals often show a strong increase in Co from core to rim. Skutterudite II strongly differs from skutterudite I, showing very high Co content coupled with the lowest Ni values ($Co_{0.49-0.59}Ni_{0.01-0.11}Fe_{0.39-0.44}$).

Di-arsenides show wide compositional ranges plotting in four main fields (Figure 6c,d):

(I)  rammelsbergite–safflorite solid solution (s.s.), with composition ranging from pure rammelsbergite to $Ram_{85.6}Saf_{11.1}$; the löllingite content is always very low ($\leq 7.6\%$);

(II)  safflorite–rammelsbergite–löllingite s.s.: such a composition is restricted to arsenides rimming—but also replacing—skutterudite, that show a relatively high safflorite content and a variable Ni/Fe ratio ($Saf_{37.4-61.9}Ram_{0.0-30.3}Löl_{21.2-62.2}$). Part of these compositions fall outside the solid solution field of [44], as also reported by other Authors (e.g., [45] with refs.).

(III)  löllingite–rammelsbergite-safflorite s.s.: all the star-shaped "safflorite" twins fall in this composition group. They are mostly characterized by low Ni content (Ram up to 17.4%, mostly close to zero) and rather wide Co/Fe ratio ($Saf_{25.3-48.2}Ram_{0-38.6}Löl_{38.3-66.5}$).

(IV)  (IV) löllingite–safflorite s.s.: this is the composition of the late-stage di-arsenides which locally enclose sulfides: Strongly enriched in Fe ($Lo_{864.7}Sa_{29.7}$ to pure Löl) and almost devoid of Ni.

Gersdorffite I is very close to the endmember composition. The much more abundant gersdorffite II shows a variable composition in the gersdorffite–cobaltite solid solution, from almost pure gersdorffite up to ca. 26% of the cobaltite endmember (Figure 6d); the Fe content is always very low.

Tetrahedrite belongs to the tetrahedrite–tennantite solid solution, its composition ranging between 75.9 and 88.9 mol% tetrahedrite end-member. Tetrahedrite II differs from tetrahedrite I for an appreciable silver content (up to 5.81 wt% Ag). Both tetrahedrite groups are characterized by irregular Hg peaks (1.26–1.74 wt%) (Table S2). Sphalerite occurs as a minor phase in the last stage, is relatively rich in Fe (ca. 8 wt.% Fe) and contains moderate amounts of Cd (0.9–1.2 wt%), within the range of the SW Sardinian sphalerites (Figure 7a,b). Absence of Hg in sphalerite is in agreement with sphalerite deposition postdating tetrahedrite, which acted as main Hg host [46].

## 4. Geological, Mineralogical, and Microchemical Features of the Co–Ni Sulfide-Bearing Cu–Ag Deposit of Piazza (Ligurian Sector of the Appennine Belt)

This section illustrates the results from preliminary investigations on the gabbro-hosted, Co–Ni-bearing, Ag-rich copper deposit exploited in the Piazza mine, inland from Deiva Marina (Eastern Liguria; Figure 11a,b).

The Piazza mineralization is probably one of the least known deposits in the outstanding cupriferous Ligurian Appennine ophiolites, east of Genova ([47,48]; see Figure 11a). In these works, the Piazza mineralization has been only briefly mentioned for due classification as stockwork-type in gabbro and compared with the Cu deposit at Campegli. However, the Piazza ore is markedly different from the pyrite–chalcopyrite–dominated assemblages recorded in Campegli and typical for most of the Ligurian volcanogenic ore deposits [49]. The Piazza mine was exploited for Cu- and Ag-rich mineralization from late 15th century until the beginning of 20th century.

The Piazza deposit is hosted along a fracture network developed in pegmatoidal gabbro and, to a lesser extent, along the contacts with some dolerite dikes crosscutting the gabbro, within the ophiolite complex of the Bracco Unit (Internal Ligurides). The ophiolite terranes of the Internal Ligurides (Figure 11b) are considered to represent a remnant of the Mesozoic Tethyan oceanic lithosphere involved in the Apennine orogenesis.

The Bracco Unit consists of pillow basalts, gabbros and serpentinite intruded by polyphase dike injections (dolerite and plagiogranite) (e.g., [50] and ref. therein), and is characterized by outstanding ophiolite–ophicalcite breccias [51] exposed in quarries. The oceanic units are topped by a thick sequence of pelagic cover units (e.g., [52]). During Tertiary these rocks suffered prehnite–pumpellyite to prehnite–zeolite facies, orogenic metamorphism mostly along main lineaments (e.g., [53,54]). Greenschist to amphibolite facies mineral assemblages, associated with both ductile and brittle structures, have been observed in gabbros and interpreted to be relics of the pre-orogenic oceanic history. This portion of the chain was affected by a multi-phase deformational history with changes from "Alpine" SW-directed to dominant "Appennine" NE-directed tectonic transport. The rock units in the Piazza–Framura sector were involved in regional-scale SW-facing F1 folds, elsewhere superimposed by large-scale NE-vergent F3 recumbent folding. Beside large-scale open folding, the Apennine tectonics produced brittle–ductile to brittle structures such as S-C structures and extensional faults and fractures, all related to gravitational collapse. Apennine deformation and low-grade metamorphism did not obliterate the primary features of the ophiolitic rocks: both intrusive and extrusive magmatic structures are still well recognizable in mafic and ultramafic rocks.

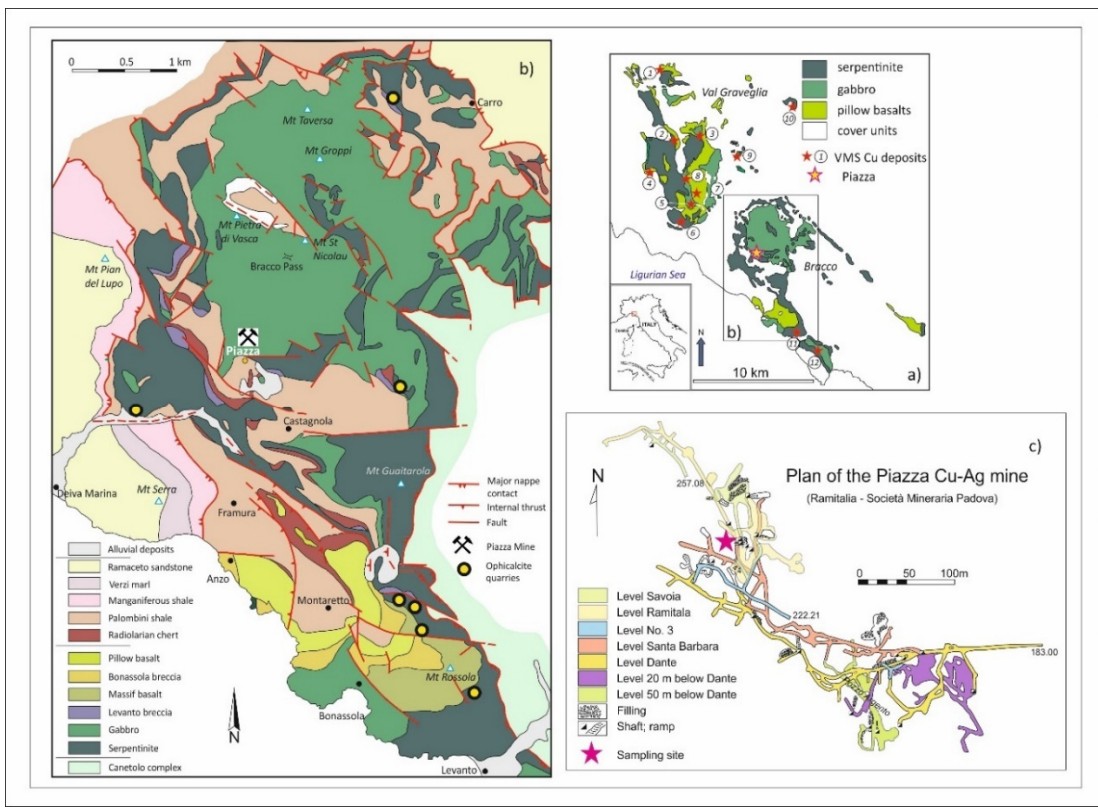

**Figure 11.** Geological setting of the Piazza mineral deposit, Northern Appenines, Eastern Liguria. (**a**) Sketchmap of the regional distribution of the ophiolite complexes in the Eastern Ligurian sector of the Northern Appennine Belt and location of the Piazza mine and of the ophiolite-hosted volcanogenic massive sulfide (VMS) deposits of the area after [47]. Location codes: (1) Reppia, (2) Monte Bianco, (3) Monte Bardeneto, (4) Libiola, (5) Campegli, (6) Casali-Monte Loreto; (7) Gallinaria, (8) Bargone, 9) Molin Cornaio; (10) Rocca di Lagorara, (11) Monte Rossola, (12) Monte Mesco. (**b**) Geological features of the Bracco ophiolite massif (after [50]) and localization of the Piazza mine and of sulfide-bearing ophicalcite quarries in [51]. (**c**) Reproduction of the historical planimetry of the Piazza mine and indication of the accessible portion (Level Ramitala) where sampling was done (after [55,56]).

## 4.1. Features of the Mineralization in the Piazza Mine

The sketch in Figure 11c shows the development of the Piazza mine workings at their utmost, during the first half of XX century, under the management of the mining company Ramitalia from Padova. Galleries were distributed in five main levels. Nowadays they are only limitedly accessible, however some noteworthy features of the ore deposit can still be observed. The original mineralization consisted of 60–80% bornite and minor chalcopyrite, included in a dominant, steep vein system with E–W trend, crosscut by minor NE–SW trending veins. According to mining reports reviewed by [55], single vein thicknesses varied between 0.5 and 1.8 m, to build up a mineralized stockwork up to 4–5 m thick. Enriched ore during production contained 16.8–45.7 wt % Cu and Ag tenors from 74 up to 480 g/ton, although no Ag phase had been reported in the past. No historical data exist about Co–Ni recovery or occurrence of the Co–Ni phases which were revealed during the present study. As noted above, the vein system was emplaced along fractures and brittle shear zones.

Cu-rich mineralized stockwork zones with an average thickness of 20 cm are still preserved on the walls and ceiling of the accessible mine galleries corresponding to the upper levels (Figure 12a,b). The vein trend may be compatible with shear extension during late-stage regional shortening in the NE–SW direction (as in [50]). Sulfide-rich mineralization typically occupies the axial part of cataclastic bands developed in the gabbro host (Figure 12c,d) and, to a lesser extent, along the gabbro–dolerite dike contact, with variable alteration of the silicate gangue [56].

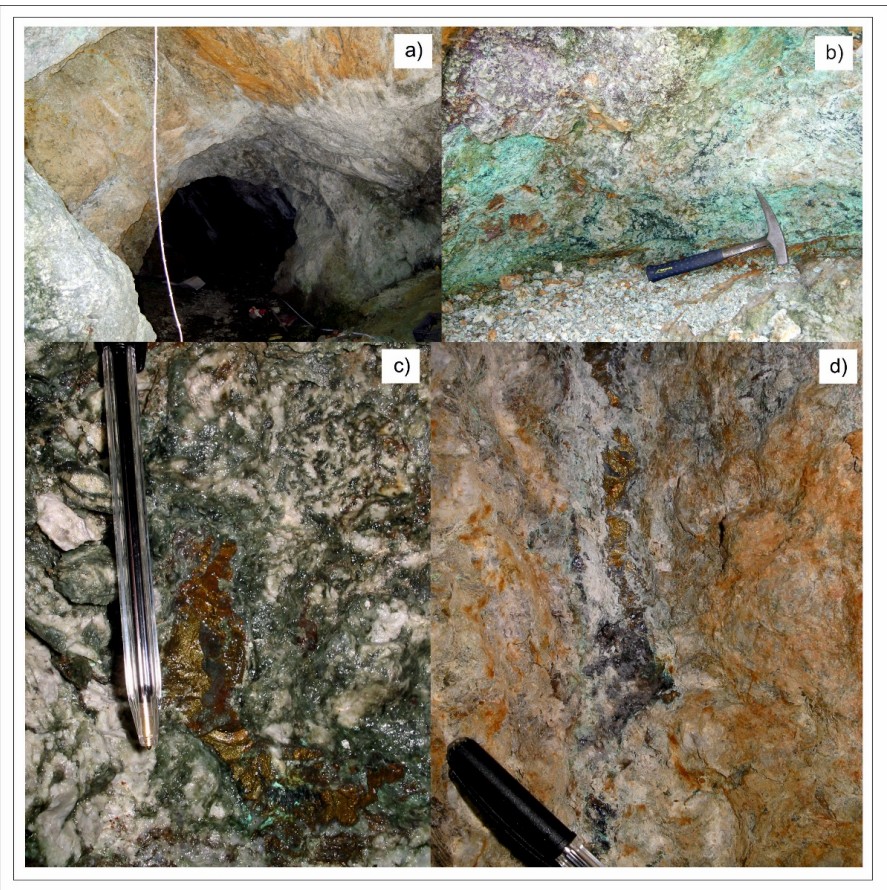

**Figure 12.** Macroscopic features of the mineralization at the Piazza mine. (**a**) Zone of hydrothermally altered, mineralized gabbro intersected by a tunnel. (**b**) Exposure of mineralized gabbro with secondary Cu minerals marking the development of the Cu-rich stockwork. (**c**) Fresh fracture of chloritized gabbro hosting irregular enrichment of Cu sulfides (chalcopyrite in orange and bornite in blue) along a thin hydrothermal veinlet. (**d**) Bornite-rich sulfide patch in whitish prehnite along veinlets in altered gabbro (the rusty color is due to surficial coatings along the mine tunnel).

Ore minerals occur as irregular aggregates and disseminated blebs within the brecciated host gabbro (Figure 13a), filling fractures and open cavities together with gangue minerals represented by prehnite, albite, minor chlorite, and extremely rare quartz. The ore is dominated by copper sulfides, bornite and chalcopyrite (the former locally altered in secondary Cu sulfides), with abundant accessory Ag-and Pb-bearing tellurides and selenides (hessite and clausthalite), Co–Ni sulfides (siegenite, Co-rich millerite), Cd-rich sphalerite, greenockite, and native tellurium. Figure 13b–o displays the typical micro-textures of the mineralization and related gangue and chlorite–prehnite alteration of the host rock. Prehnite is the main gangue phase. It occurs as fine- to coarse-grained, zoned euhedral crystals, the latter in open cavities often intergrown with or lined by thin selvedges of albite crystals (Figure 13b–d). Chlorite is a common alteration product especially near the mineralized veinlets (Figure 13a,b), where it also occurs in felt-, rosette-like, and vermicular aggregates intergrown with fine-grained prehnite (Figure 13e). The fractured host gabbro shows the most intense evidence of deep interaction with the mineralizing fluids. Away from the mineralized vein stockwork the post-magmatic assemblages partly replacing the primary minerals in both gabbro and dolerite (and probably related to oceanic metamorphism) include amphiboles (hornblende, actinolite, and tremolite), albite, epidote and chlorite. Epidote was never observed in the gangue and vein-related alteration assemblage of the mineralized samples examined.

Ore textures are dominated by bornite–chalcopyrite intergrowths in the irregular, nodular to net-textured patches along the axis of the veinlets (Figure 12c,d). Samples show variations in relative abundance of the main Cu sulfides. Bornite commonly occurs as small to large, net-textured aggregates intergrown with chalcopyrite and the accessory phases and enveloping gangue crystals (Figure 13f,g). Bornite commonly shows extremely fine-grained, sub-micrometric spindle-like chalcopyrite intergrowths and is locally altered into secondary covellite–idaite (Figure 13f,g–l). Both chalcopyrite and bornite are inclusion-rich (Figure 13i–l) especially towards the margins of the veinlets. White to cream-white subhedral crystals of Co–Ni sulfides siegenite and millerite (Figure 13g,h,m) are variably disseminated in the Cu sulfide aggregates and frequently along the margins of the large cupriferous aggregates in contact with the prehnite crystals. Sphalerite occurs in accessory amounts in association with chalcopyrite–bornite (Figure 13i) and with irregular distribution, mostly as fine-grained amoeboidal–vermicular blebs occasionally intergrown with hessite, and, only locally, as coarse-grained, anhedral patches rich in cloud-like disseminations of very fine-grained sulfosalts (Figure 13n). Characterized by fine grain size, Ag and Pb selenides–tellurides (hessite–naumannite, clausthalite–altaite) are rather common and occur as amoeboidal blebs either isolated or intergrown to one another (Figure 13i,j). The sulfosalts are variably disseminated in the chalcopyrite + bornite aggregates (Figure 13k,l). Hessite might be the main Ag carrier in the Piazza mine. Hessite and clausthalite (sub)micro-inclusions are also observed in siegenite, millerite and sphalerite (Figure 13m,n). Accessory native tellurium and greenockite (Figure 13o) were detected by SEM inspection during microprobe analysis: a subhedral greenockite crystal was observed between chalcopyrite and prehnite, while micrometric grains of native Te were detected in chalcopyrite. Figure 3c proposes a paragenetic scheme for the accessible part of the Piazza mineralization.

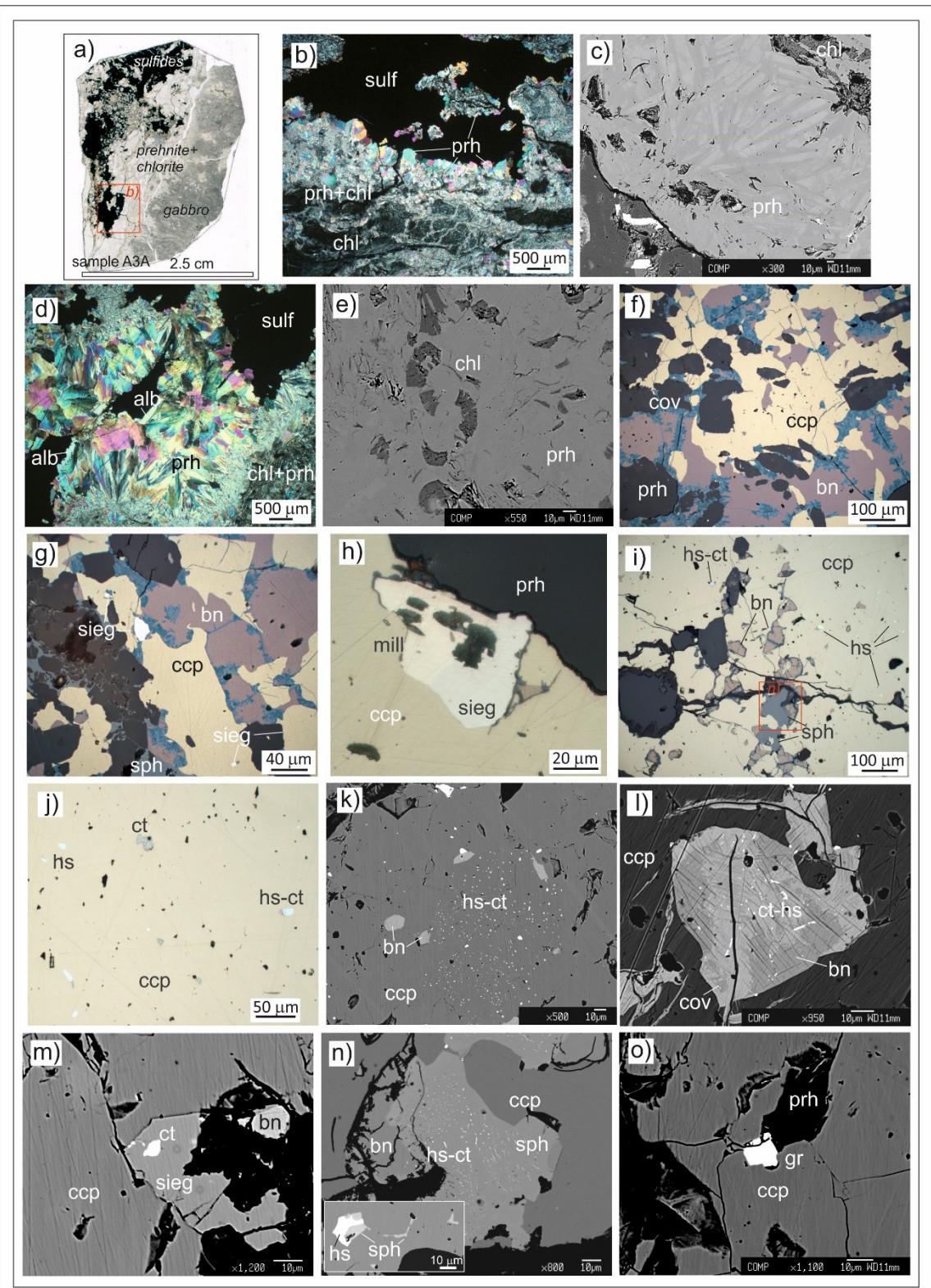

**Figure 13.** Microscopic features of the mineralization from the Piazza deposit. (**a**) Scan image of a thin section showing the distribution of sulfides (black), ore-related gangue and prehnite–chlorite alteration (pale yellowish grey) in mildly chloritized gabbro (dark grey-greenish): (**b**) Particular of the mineralized veinlet in (**a**) with sulfide aggregate, euhedral prehnite crystals and prehnite–chlorite intergrowths enveloping deeply chloritized fragments of host gabbro. (**c**) Mild Fe zoning in hydrothermal euhedral

prehnite. (**d**) Sulfide patch intergrown with euhedral prehnite associated with albite as anhedral rims. (**e**) Intergrowths of fine-grained prehnite (mid grey) and chlorite lamellae (dark grey) in vermicular to booklet-like aggregates. (**f**) Bornite–chalcopyrite aggregate enveloping, and corroding, prehnite crystals, and with mild replacement by secondary covellite–idaite along cracks. (**g**) Bornite–chalcopyrite aggregate enclosing white siegenite grains. (**h**) Composite Co–Ni sulfide grain along the chalcopyrite–prehnite contact and consisting of siegenite and millerite. (**i**) Panoramic view of a large chalcopyrite–bornite aggregate rich in disseminated hessite and clausthalite and including anhedral sphalerite blebs. (**j**) Close-up view of some hessite and clausthalite micro-inclusions in chalcopyrite. (**k**) Cloud of hessite–clausthalite (sub-) micro-inclusions affecting chalcopyrite with remnants of bornite. (**l**) Bornite bleb with thin spindle-like chalcopyrite lamellae (black), altered by covellite along cracks and rich in (sub-) micro-inclusions of clausthalite ± hessite. (**m**) Clausthalite inclusion in siegenite along the chalcopyrite–prehnite contact. (**n**) Close-up view of one of the anhedral sphalerite blebs in (**j**) showing extremely fine-grained dissemination of Se–Te sulfosalts and, in the inset, hessite intergrown with sphalerite in chalcopyrite. (**o**) Subhedral crystal of greenockite grown at the chalcopyrite–prehnite contact. (b, d: plane polarized transmitted light; f, g, h, i, j: plane polarized reflected light; c, e, k, l, m, n, o: electron microscopy backscattered images). Abbreviations: alb = albite, bn = bornite, chl = chlorite, cov = covellite, cpy = chalcopyrite, ct = clausthalite, gr = greenockite, hs = hessite, mill = millerite, prh = prehnite, sieg = siegenite, sph = sphalerite, sulf = sulfides.

## 4.2. Chemical Composition of Ore and Gangue Minerals at Piazza

Table S3 contains representative chemical compositions of ore and gangue minerals in the Piazza vein stockwork. Bornite displays variable composition mainly depending on how fine-grained the mutual intergrowth with chalcopyrite is. Bornite contains small quantities of Ag (mostly below 0.1 wt%), further enriched in alteration to idaite and covellite (up to 0.3 wt%). Chalcopyrite is in general almost pure. Sphalerite is relatively low in Fe (maximum 1.57 wt%) and often very Cd-rich, although the Cd content is highly variable and locally outstanding, ranging between 2.5 wt % to over 9 wt % Cd (Figure 7a,b), well above the SW Sardinian and Usseglio sphalerites. As shown in Table S3, detectable accessory elements in sphalerite are represented by Pb (0.4–0.14 wt %), Se (0.06–0.6 wt %; Figure 7d), Co (0.046–0.14 wt %; Figure 7c), Ge (0.02–0.04 wt %; Figure 7c) and, occasionally, Te (max. 0.1 wt %) and Ni (max. 0.07 wt%). The Cu, Pb, Te, Se, and Ag impurities might be related to the Cu sulfide host and telluride–selenide micro-inclusions. The Co–Ni sulfides at Piazza display compositions and stoichiometries corresponding to siegenite (Co, Ni)$_3$S$_4$ and cobaltiferous millerite (Ni,Co)S (Table S3). Siegenite shows variable Co/Ni ratio, from 1.54 to 0.98, and always contains minor amounts of Fe which are minimum (below 1 wt %) in the Co-richest zones and increase (up to 6.5 wt %) where Co contents are lower (23 wt%). Millerite displays a rather constant accessory Co, between 3.79 and 5 wt %, and irregular Fe contents, between 0.2 and 5.7 wt%. Both (inclusion-free) siegenite and millerite always contain detectable impurities of Se (max. 0.18 in both millerite and siegenite) and Te (avg 0.21 wt% in siegenite; 0.47 wt % in millerite), although not dependent from rare Ag and Pb impurities.

Sulfosalts are characterized by (a) sub-micrometric textural and compositional inhomogeneities probably related to mutual intergrowths, contamination from with the host phases and possible internal zoning at (sub-)micron scale. The main Ag-phase is represented by hessite, Ag$_2$Te, containing highly variable Se (towards the naumannite term, Ag$_2$Se), frequent accessory Cd (up to 0.4 wt %) not directly related to presence of Zn impurities from sphalerite, and occasional Sb contents (max 0.08 wt%). Clausthalite, PbSe, micrograins display variable composition, with excess S, Cu, and Fe derived from the host copper sulfides and with Te up to 3.6 wt %. Tellurium contents in clausthalite may only partly depend on intergrowth with hessite grains at submicrometric scale: Te abundance does not necessarily correlate with occasional Ag contents (up to 3.8 wt %), thereby suggesting remnants of earlier altaite (PbTe) [57]. Alternatively, some Te peaks (coupled with detectable Sb) may correspond to intersection of sub-micro-inclusions of native tellurium. Ag-bearing clausthalite blebs are also Bi-bearing (up to 1.7 wt %). The rare, micrometric grains of native Te detected in chalcopyrite, and amenable for analysis, are partly contaminated by the sulfide host, although they contain measurable amounts of Se, Sb, Pb,

Ag, and Co (Table S3). Beside the outstanding Cd presence in sphalerite and, to a lesser extent, in hessite, the cadmiferous character of the mineralization is revealed also by the detection of micrometric greenockite crystals intergrown with chalcopyrite and prehnite (Figure 13o).

Prehnite occurs as major gangue phase, as alteration-related mineral, as well as in the mafic country rocks away from the mineralization (Table S3). Prehnite displays minor compositional variability mainly involving Fe and Al. Ore-related prehnite at Piazza is generally Fe-poor (if compared with data in [58]), although some Fe zoning is visualized by in the euhedral prehnite crystals within the mineralized veins (Figure 13c), with Fe-richer cores overgrown by the Fe-poor rims. Prehnite in deeply altered gabbro wallrock tends to show slightly higher Al contents.

Vein-hosted albite is basically pure, with lower CaO (below 1 wt %) and very low Fe impurities (below 0.12 wt %) compared with metamorphic albite in the host rocks.

Chlorite varies in its interference colors, textures and, to a lesser extent, composition according to its occurrences, i.e., as alteration phase of cataclastic gabbro at the margin of the ore veins and as ore vein filling with lamellar prehnite, or else variably replacing primary mafic minerals (pyroxene, hornblende) as felts with prehnite and actinolite/tremolite. The majority of data on chlorite were obtained from crystals from the ore zone. In the classification diagram in Figure 14a, most of the chlorite data from the mineralization plot in the low parts of the picnochlorite–diabantite fields, while chlorite in unmineralized gabbro is characterized by further lower Fe contents. All chlorite data are representative of Mg-rich varieties, with low Cr contents. When plotting $Al^{IV}$ versus #Fe (Figure 14b), the Piazza chlorites plot in concordance with the field for hydrothermal compositions.

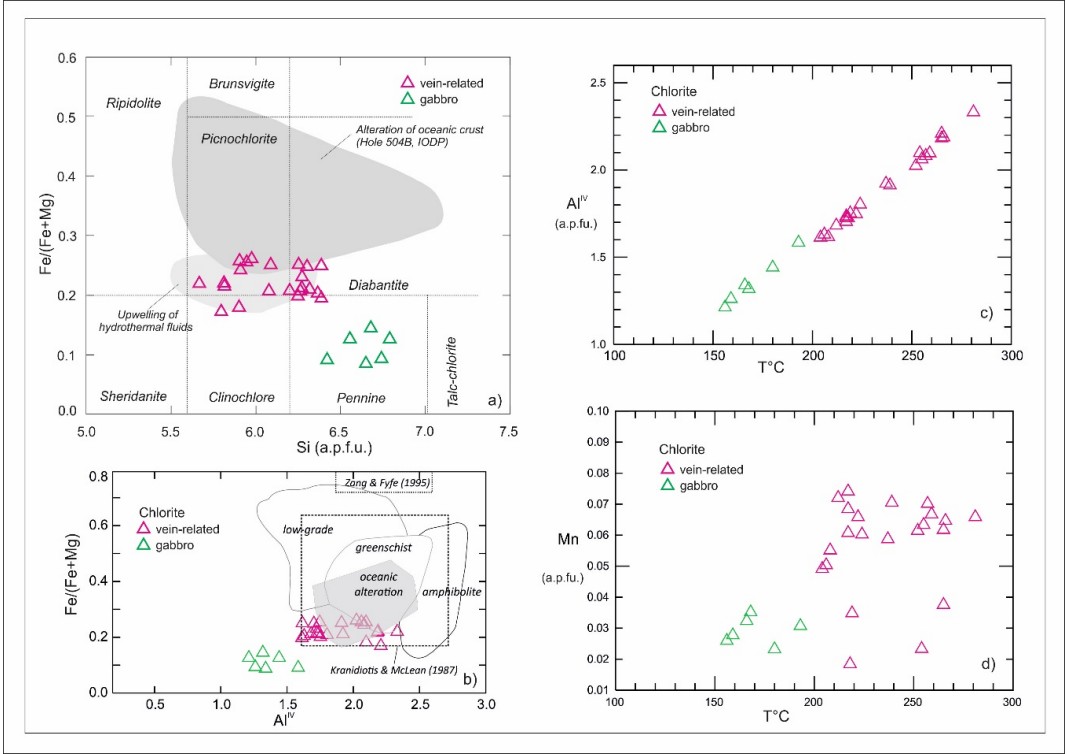

**Figure 14.** Chlorite composition and geothermometric estimate. (**a**) Chlorite compositional data, distinguished between gabbro hosted (far from ore) and vein-related, plotted in the classification scheme by [59]. Fields for chlorites: [60] for altered oceanic crust (Hole 504B, IODP), [61] from alteration related to upwelling of hydrothermal fluids (Mid-Atlantic Ridge, Galapagos, East Pacific Rise 13° N, Izu-Bonin). (**b**) Plot of $Al^{IV}$ vs Fe/(Fe+Mg) data of Piazza chlorites for comparison with oceanic alteration-related and metamorphic data after [61] and for evaluating the applicability of chlorite geothermometers [62,63]. (**c**) Plot of temperature vs $Al^{IV}$ for the Piazza chlorites according to the chlorite geothermometer proposed by [62]. (**d**) Plot of temperature vs Mn content for the Piazza chlorites.

## 5. Geothermometric Estimates and Fluids

### 5.1. Fluid Inclusion Analyses on the Usseglio Veins

Fluid inclusions were analyzed in mineral phases related to the siderite, baryte and Co–Ni–Fe arsenides stage.

Siderite stage—Due to weathering, siderite I is not suitable for a fluid inclusions study. Only rarely, euhedral quartz crystals associated with siderite I contain fluid inclusions along healed microfractures. Fluid inclusions along trails that do not reach grain boundaries could be pseudosecondary, though petrographic observations do not allow to unequivocally recognize if they are of secondary or pseudosecondary origin. They are two-phase (liquid + vapor), aqueous liquid-rich inclusions (LV), with irregular shape with a relatively high degree of filling (ca. 0.9). Inclusions of surely secondary origin do occur, and have not been considered.

Baryte stage—Fluid inclusions in baryte (and, subordinately, quartz) are abundant. On the basis of the phase assemblage at room temperature, the following types of fluid inclusions can be distinguished: two- or three-phase (liquid + vapor ± solid), liquid-rich inclusions (LV and LVS, respectively), and two-phase (vapor + liquid) vapor-rich inclusions (VL) (Figure 15a). About 70% of the liquid-rich inclusions contains a tiny birifrangent crystal that has been identified as calcite by micro-Raman spectroscopy. The petrographic observations clearly show that LV/LVS and VL inclusion types were simultaneously trapped within single fluid inclusion assemblages (FIA of [62]). In particular, they often occur in the same cluster and/or growth zone, as primary inclusions, according to criteria by [63]. A strong variation of liquid/vapor ratio is observed, from LV/LVS to VL inclusions, suggesting heterogeneous trapping processes (i.e., the contemporaneous trapping of the volatile phase together with the liquid phase in single inclusions) during a boiling event. Coexisting inclusions with strongly different liquid/vapor ratio are also common along healed microfractures.

Co–Ni–Fe arsenides stage—Quartz and siderite of this stage rarely contain primary fluid inclusions aligned along growth zones (Figure 15b), as well as isolated inclusions of possible primary origin. Primary fluid inclusions along growth zones in quartz mostly display irregular shape, while those in the siderite (both along growth zones and isolated) mainly show a negative crystal shape. The fluid inclusions are always two-phase (liquid + vapor), liquid-rich (LV), with a consistent degree of filling (ca. 0.9).

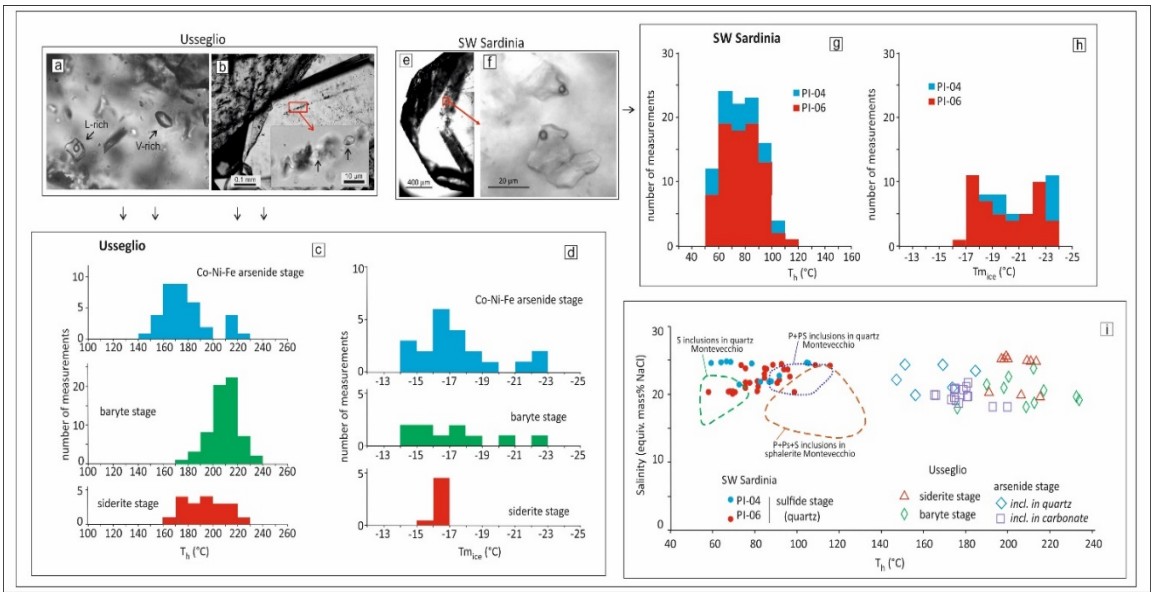

**Figure 15.** Photomicrographs and frequency histograms of fluid inclusions from the Usseglio (a to d) and SW Sardinia vein systems (e to h), and Th vs. salinity diagram for both vein systems. Usseglio:

(**a**) Baryte stage: coexisting liquid-rich and vapor-rich fluid inclusions. The liquid-rich inclusion also contains a tiny crystal of calcite. (**b**) Co–Ni–Fe arsenides stage: quartz crystal containing alignments of primary liquid-rich fluid inclusions (enlarged within the inset) (transmitted light, parallel nicols). (**c**) and (**d**): Frequency histograms of (**c**) Homogenization temperature ($T_h$) and (**d**) final ice melting temperature ($Tm_{ice}$). SW Sardinia: (**e**) low-magnification photomicrograph of a quartz crystal from the Ni–Co mineralization showing a large growth zone delineated by abundant fluid inclusions (not visible at this magnification). (**f**) Enlargement of the red square inset of the previous photomicrograph showing two relatively large primary two-phase (LV) inclusions (transmitted light, parallel nicols). (**g**), (**h**): Frequency histograms of (**g**) homogenization temperature ($T_h$) and (**h**) final ice melting temperature ($Tm_{ice}$). (**i**) Homogenization temperature ($T_h$) vs. salinity diagram for the Usseglio and SW Sardinian vein systems. The data of the SW Sardinian vein system are compared with the $T_h$/salinity fields of the fluid inclusions examined in quartz and sphalerite of the Pb–Zn–Ag Montevecchio deposit by [30]. Abbreviations: P = primary, S = secondary, PS = pseudosecondary inclusions.

Microthermometric Results

Homogenization temperature ($T_h$), hydrohalite melting temperature ($Tm_{hh}$), and final ice melting temperature ($Tm_{ice}$) results are summarized in Table 1 (details in Table S4a) and shown as frequency histograms in Figure 15c,d; the $T_h$-salinity relation is shown in the diagram of Figure 15i (right side). In general, LV and LVS inclusions are characterized by very low temperature of initial melting: the appearance of the "orange peel" texture, which is typically observed 5–10 degrees below the eutectic temperature [62] generally occurs between −55 and −45 °C, and in the biggest inclusions initial melting is observed at ca. −52 °C. No melting of solid $CO_2$ or clathrate was observed. These observations suggest that the fluid composition can be broadly modeled in the H2O–NaCl–CaCl2 system [64]. The main characters of the fluid inclusions for the different stages (earlier to later) are summarized below.

Siderite stage—During low temperature experiments only part of the inclusions froze even going down to ca. −190 °C. Once frozen, with increasing T initial melting is seen, in the biggest inclusions, at ca. −50 °C. At ca. −25 °C the inclusions consist of vapor, liquid and three solids: ice, hydrohalite and an unknown phase (typically greenish and with elongated shape). Between −23.7 and −22.5 °C strong melting occurs with disappearance of hydrohalite ($Tm_{hh}$). With subsequent heating ice melts between −17.0 and −16.0 °C and the greenish solid phase finally melts between +14.0 and +18.4 °C. Assuming entrapment of a H2O–NaCl–CaCl2 fluid, the weight fractions of NaCl and $CaCl_2$, calculated by using $Tm_{hh}$ and $Tm_{ice}$ pairs according to [65], are 12.2–16.0 and 3.4–7.2 mass %, respectively. Such data are a first approximation, as the occurrence of the greenish solid phase melting at relatively high temperature suggests that the system was more complex. The similarities with the microthermometric behavior of sulfate brines inclusions reported by [66] suggest that the final phase to melt may be a sulfate, in agreement with the precipitation of abundant baryte during the subsequent stage. On heating, total homogenization to the liquid phase occurs between 168.9 and 221.1 °C.

Baryte stage—Due to the entrapment of a heterogeneous fluid, the fluid inclusions show strongly different microthermometric behavior. As the determination of the temperature of phase transitions (particularly, $T_h$) in the VL inclusions is affected by a huge error, the study was performed on the liquid-richest LV inclusions, that likely represent entrapment of the liquid phase only [67]. Once frozen, in these inclusions initial melting is (rarely) seen at ca. −52 °C, and some melting is evident above −45 °C. Melting continues until, between −24.2 and −22.7 °C, hydrohalite disappears. Ice melting is generally observed at a slightly higher temperature (−22.1–15.1 °C). The calculated weight fractions of NaCl and $CaCl_2$ are 11.8–18.8 and 4.6–6.5 mass%, respectively. On heating, homogenization into the liquid phase occurs between 176.0 and 234 °C. In LVS inclusions, calcite is still present when the vapor disappears.

Co–Ni–Fe arsenides stage—Also in this case the biggest LV fluid inclusions of this stage show initial melting at very low T (ca. −52 °C); with further heating hydrohalite melts at temperature between −25.1 and −21.8 °C, and ice between −23.1 and −14.4 °C. These $Tm_{hh}$–$Tm_{ice}$ pairs correspond to calculated

weight fractions of NaCl and $CaCl_2$ of 11.2–18.2 and 1.7–6.6 mass%, respectively. The homogenization temperature (into the liquid) ranges from 147.3 to 220.8 °C. While the measurements in quartz fall between 147.3 and 173.7 °C (with an outlier at 184.7 °C), those in siderite are shifted to higher temperatures (156.8 to 220.8 °C), possibly due to post-entrapment modifications.

**Table 1.** Summary of final ice melting temperature ($Tm_{ice}$), homogenization temperatures ($Th$), hydrohalite melting temperature ($Tm_{hh}$) data and computed salinity for primary (P) and pseudosecondary–secondary (PS–S) fluid inclusions from the Usseglio and SW Sardinian vein systems. Number of measurements are reported within brackets.

| Sample | Stage | Host Mineral | F.I. Origin | Tm$_{hh}$ Range (°C) | Tm$_{ice}$ Range (°C) | T$_h$ Range (°C) | Salinity (eq. Mass wt% NaCl) | Salinity (Mass% NaCl + CaCl$_2$) |
|---|---|---|---|---|---|---|---|---|
| | | | | *Usseglio Veins System* | | | | |
| US110 | siderite | quartz | PS-S | −23.7 to −22.5 [10] | −17.0 to −16.0 [10] | 168.9 to 221.1 [19] | 19.5 to 20.2 | 19.0 to 19.8 |
| OF2982 | baryte | baryte | P + PS-S | −24.2 to −22.7 [6] | −22.1 to −14.1 [10] | 176.0 to 234.0 [63] | 17.9 to 23.8 | 18.3 to 23.5 |
| Bar1 | Co-Ni-Fe | quartz | P | −23.9 to −23.1 [13] | −23.1 to −15.0 [8] | 147.3 to 184.7 [10] | 18.6 to 24.4 | 18.2 to 24.0 |
| Bar1 | Co-Ni-Fe | siderite | P | −25.1 to −21.8 [17] | −19.0 to −14.4 [15] | 156.8 to 220.8 [26] | 18.1 to 21.7 | 17.8 to 21.2 |
| | | | | *SW Sardinian Vein System* | | | | |
| PI-04 | | quartz | P+PS-S | | −23.6 to −18.6 [12] | 52.4 to 105.2 [22] | 21.4 to 24.8 | |
| PI-06 | | quartz | PS-S | −22.5 [1] | −23.1 to −16.8 [32] | 54.5 to 115.7 | 20.1 to 24.4 | 20.0 |

## 5.2. Fluid Inclusion Analyses on the SW Sardinian Veins

Two samples from Pira Inferida mine (PI-04 and PI-06) were selected for fluid inclusions analyses because of the relative higher abundance of inclusions compared to samples from the other veins. Fluid inclusions were studied in quartz-rich samples, because Fe carbonates were often darkened by incipient oxidation. In these samples, euhedral and sub-euhedral quartz crystals are often characterized by growth zones defined by fluid inclusions of primary origin according to [63] (Figure 15e). In general, fluid inclusions within growth zones are of small size (usually <10 μm), although in few growth zones relatively large (up to 80 μm) inclusions, often characterized by irregular shape, were observed (Figure 15f). Fluid inclusions (from <10 to 50 μm), also occur along trails associated to healed fractures; petrographic observations do not allow to recognize if they are secondary or pseudosecondary inclusions. Two fluid inclusion types can be distinguished on the basis on the phase assemblage present at room temperature: i) two-phase (liquid + vapor), liquid-rich, aqueous fluid inclusions (LV) characterized by a high degree of fill (ca. 0.95 or higher) and ii) single-phase (liquid), aqueous fluid inclusions (L). Both L and LV inclusions often coexist in growth zones and in healed fractures (i.e., within single FIA). In some cases, L inclusions can be rather abundant and can constitute the majority of the fluid inclusions of a single FIA. The absence of a vapor phase in L inclusions likely resulted from failure of bubble nucleation due to metastability processes, which usually characterize aqueous fluid inclusions with very high fluid density [62,63].

To promote vapor bubble nucleation in L inclusions before microthermometric analyses, the double polished section prepared for fluid inclusion studies were kept for two weeks at low temperature ca. −18° C) within a freezer, according to [62]. This procedure led to the formation of bubbles in a limited number of L inclusions which, therefore, become LV inclusions.

Microthermometric Results

Microthermometric analyses were carried out on primary and pseudosecondary–secondary LV inclusions. Since the T$_h$ and Tm$_{ice}$ ranges of primary and pseudosecondary–secondary inclusions

are similar their data were merged together in Table 1 (details in Table S4) and Figure 15g,h. Salinity ranges are also reported in Table 1, whereas the $T_h$ and salinity data are shown in binary diagram of Figure 15i (left side). The diagram also displays, for comparison, the $T_h$-salinity fields of fluid inclusion from the nearby Montevecchio Pb–Zn–Ag deposit studied by [30]. Final ice melting temperature could not be measured in all inclusions for which $T_h$ was detected. In fact, in several LV inclusions in which the bubble disappeared upon freezing non-equilibrium ice melting occurred (c.f. [63]). On the other hand, in some L inclusions a bubble nucleated upon freezing and at least $T_h$ could be measured in inclusions. During low-temperature experiments, after solidification, the inclusions are filled by micro-crystals. Then, during heating a progressive coarsening of the micro-crystals occurred and first melting could be observed in the larger inclusions between about −50 and −45 °C, close to the eutectic temperature of the H2O–NaCl–CaCl2 system, suggesting that the trapped fluid contains dissolved $CaCl_2$ and NaCl [64,68,69]. On subsequent heating final ice melting occurred between −23.9 and −16.8 °C. In particular, a rough bimodal distribution of $Tm_{ice}$ is shown by Figure 15h. Salinity computed from $Tm_{ice}$ by the use of the equation of [70] varies from 20.1 to 24.6 equiv. mass% NaCl (Table 1). In a single relatively large inclusion final melting of hydrohalite ($Tm_{hh}$) could be observed at −22.5 °C, whereas ice totally melted at −17.2 °C in this inclusion. Assuming that this inclusion trapped a $CaCl_2$–NaCl-bearing aqueous fluid, the NaCl and $CaCl_2$ contents, computed from $Tm_{hh}$ and $Tm_{ice}$ and the model of [65], are 16.5 and 3.5 mass%, respectively. Thus, the total salinity ($NaCl+CaCl_2$) is 20 mass%, which is very close to the salinity computed only from $Tm_{ice}$ (i.e., 20.4 equiv. mass% NaCl). Clathrate was not observed during low-temperature analyses. On heating, fluid inclusions showed total homogenization (into the liquid phase) between 52.4 and 115.7 °C, with most of the $T_h$ values between 60 and 90 °C (Figure 15g).

*5.3. Chlorite Geothermometer for the Piazza Cu Mineralization*

Chlorite composition is considered to be temperature- (and pressure-) dependent and has been employed for empirical geothermometers frequently applied to hydrothermal/geothermal contexts. [71,72] reviewed chlorite geothermometry based on dated and recent empirical, semi-empirical and thermodynamic methods and stressed the difficulties deriving from a series of factors including oxidation state of Fe and vacancies especially regarding low-temperature metamorphic/diagenetic chlorites. Most of the chlorite data considered here for Piazza are texturally related to the hydrothermal mineralizing process. Although the ore-related chlorites show a characteristic increase in Fe with $Al^{IV}$ (Figure 14b), they display relatively low Fe contents. In the chlorite stoichiometry, calculated with Norm program, all Fe occurs as bivalent Fe. An accurate and proper study on chlorite crystal chemistry is, however, beyond the scope of this work. The aim of our data is just to obtain a reasonable temperature estimate by applying one of the semi-empirical methods employed in studies on hydrothermal deposits. We applied the method by [73], which is based on $Al^{IV}$ contents but, like the method by [74], also applies corrections regarding the positive dependence between temperature and Fe in octahedral coordination. Both methods were considered able to provide realistic results when compared with fluid inclusion data on ore deposits (e.g., in [75]). The method by [69] employs the function $T(°C) = 106\ Al^{IV}_C + 18$, where $Al^{IV}_C = Al^{IV} + 0.7\ [Fe/(Fe + Mg)]$. The choice of this method, instead of the one by [74], was advised by [76] for Fe-poor hydrothermal chlorites. The resulting temperature estimates (reported in Table S3) are plotted in the diagram in Figure 14c. The range of temperatures for the vein-related chlorites at Piazza is comprised between 200° and 280 °C (average 236 °C). These results are compatible with the ore-related hydrothermal assemblage lacking Fe-rich epidote, which would be indicative of temperatures exceeding 300 °C [77]. All the analyzed chlorites contain measurable Mn (Figure 14d) and low to very low Cr (Table S3), in agreement with the basaltic nature of the host rocks [49]. However, a tendency is seen for the ore-related chlorites to enrich in Mn.

## 6. Carbon and Oxygen Isotope Data on Ore-Related Carbonates

The isotopic, $\delta^{13}$C and $\delta^{18}$O composition of the siderite gangue in representative samples from the SW Sardinian (Monte Linas) and Usseglio Ni–Co-rich vein systems is reported in Table S5. Data are plotted in Figure 16a,b. The Usseglio data are representative of both Co–Ni-rich (siderite III: Punta Corna) and Co–Ni-poor (siderite I: Taglio del Ferro) orebodies across the whole vein system. The data for the SW Sardinian veins belong to samples from the orebodies of the Sa Menga and Acqua Is Prunas veins where Fe carbonate gangue was much less oxidized than in the Pira Inferida samples. In the $\delta^{18}$O-SMOW versus $\delta^{13}$C-PDB diagram (Figure 16a) the SW Sardinian data are spread in a roughly elliptical cloud with moderately light $\delta^{13}$C between −2.9 and −7.5‰ and $\delta^{18}$O between 13.6 and 18.2‰, while the Usseglio samples plot in a very restricted field with $\delta^{13}$C at −7‰ and $\delta^{18}$O between 17.3 and 18.7‰, with minimal differences between the Co–Ni-rich and the Co–Ni-poor veins.

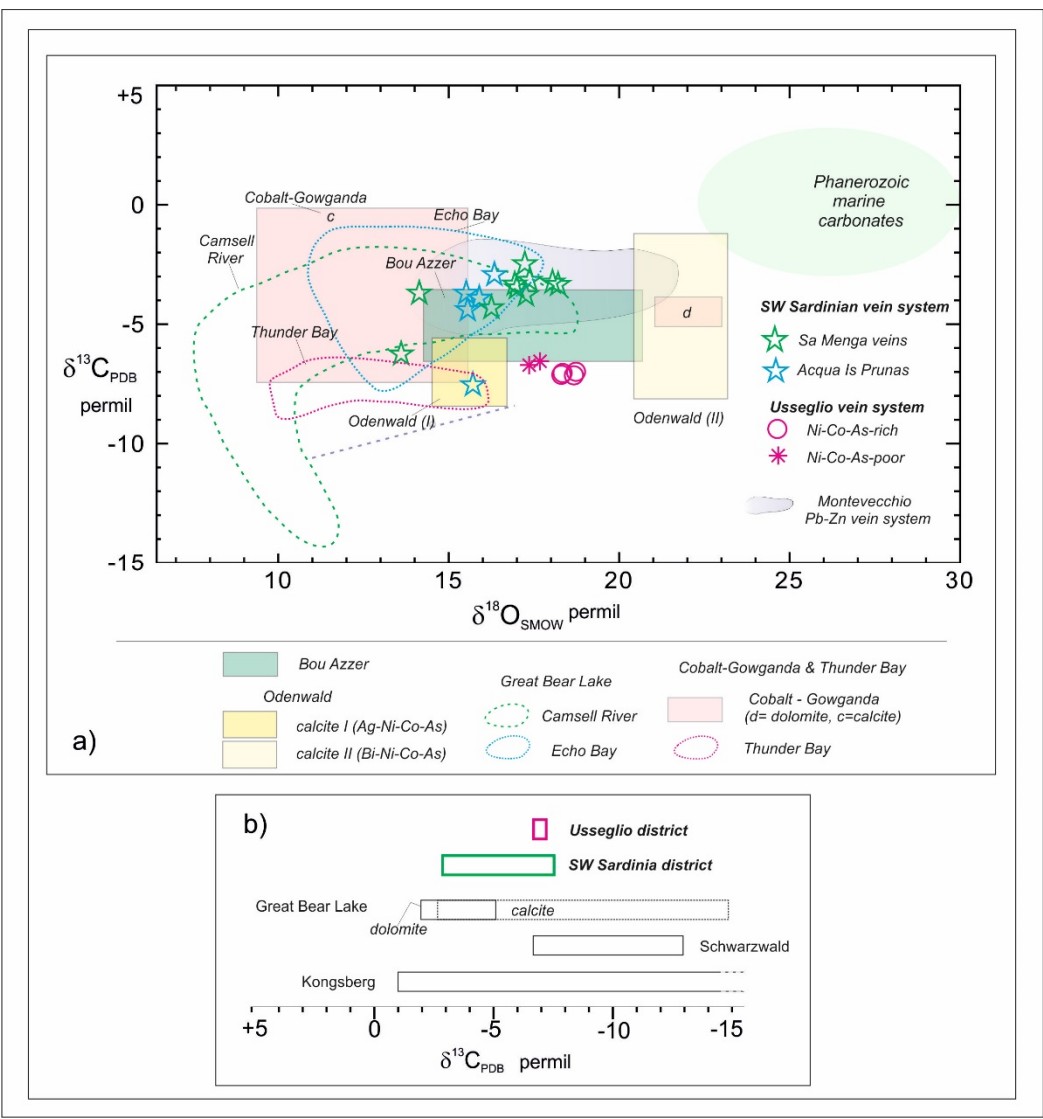

**Figure 16.** Stable isotope composition of ore-related carbonates. (**a**) Carbon and oxygen and (**b**) carbon-only isotopic composition of Fe carbonates from the SW Sardinian and Usseglio vein systems and comparison with isotopic data from the Montevecchio Zn–Pb vein system [30] and from several Co–Ni-rich, five element vein-type ore deposits world-wide [1,7,78–84]. Compositional field for Phanerozoic marine limestones after [78].

Additional data fields shown for comparison depict the range of the isotopic signatures of marine carbonates of Phanerozoic age [78], and of carbonate gangue (siderite, dolomite, calcite) from several worldwide Ni–Co bearing, five-element-vein-type mining districts, i.e., Bou Azzer (Maroc; dolomite and calcite, [79]), Odenwald (SW Germany; calcite, [7]), Cobalt-Gowganda, Thunder Bay and Great Bear Lake (Canada; [1,80–82]). In Figure 16b the $\delta^{13}$C values of the SW Sardinian and Usseglio vein systems are plotted together with additional dataset from reference five element vein-type districts worldwide from literature, like Schwarzwald (Germany; [83]), Kongsberg (Norway; [1,84] and Great Bear Lake (Canada, with distinction between dolomite and calcite signatures). For further comparison, a data cloud is added with signatures of siderite from the Montevecchio Pb–Zn–Ag vein system [30] sharing the geographical and geological setting as well as thermal and fluid conditions (low temperature brines) with the SW Sardinian one. The data ranges of the SW Sardinian and Usseglio vein systems display some distinct yet characteristic signatures in common with various other five element vein-type ore districts. The expanded SW Sardinian field is contrasting with the strongly restricted range of the Usseglio vein system, although the two data domains broadly overlap. At Usseglio minimal differences are recorded between the Co–Ni-rich veins of the eastern sector of the mining area and the siderite-only veins occurring in the central and western sector. The data for Usseglio are preliminary, but such similarity in both $\delta^{13}$C and $\delta^{18}$O compositions suggests, in agreement with the field and petrographic data, that the two vein types represent different assemblages of the same ore system. As shown in Figure 16a,b, the dimension of data fields as well as the carbonate species involved in the ore-related gangue do not seem to be discriminant features in the five-element vein-type ore systems worldwide. The reference data fields represent isotope signatures for different carbonates (dolomite, calcite, siderite) with distinction, in some cases, between distinct ore stages.

The isotopic patterns appear to be quite variable and suggestive of fluid evolution or, else, fluctuations in physico–chemical parameters during multistage ore deposition, with or without involvement of different carbonate species. However at least one feature is in common to all ore districts, i.e., the mildly to moderately negative $\delta^{13}$C signatures of ore-related carbonates. Hence the Sardinian and Usseglio data are fully compatible with the reference ore systems. Interestingly, the Usseglio data plot very close to the $\delta^{13}$C-restricted field of ophiolite-hosted Bou Azzer. The SW Sardinian data partly overlap with the Montevecchio data cloud, although they tend to spread their $\delta^{13}$C values towards the more negative range of compositions, thereby resembling the wide fields of the Canadian deposits.

## 7. Discussion

The Ni–Co-rich vein systems in SW Sardinia and near Usseglio in the Western Alps display distinct geological settings, and probably age, but share many characteristics regarding style of mineralization and, above all, mineral assemblages. The better exposed Sardinian veins allow to reconstruct a wider paragenetic scheme where the scheme for the Usseglio vein system can perfectly fit. Moreover, in both vein systems textures and mineral assemblages for both metallic and non-metallic phases are largely compatible with what reported for several of the five-element vein-type deposits worldwide in old as well as recent literature. The most characteristic features are represented by the arsenide-dominated ore assemblages, the gangue assemblages (with carbonates, quartz +/– baryte), the paragenetic scheme (comprising a Ni–Co-dominated stage followed by deposition of base metal sulfides + tetrahedrite) and the endowment of accessory components (Bi, Ag, etc.), variably distributed in the orebodies. The arsenide assemblages (Figure 6a,b) depict a more multifaceted crystallization sequence for the SW Sardinian veins, where the progression from mono- to di- to tri-arsenides and/or sulfarsenides, variably enriched in Sb, may reflect fluctuations in As and Sb abundances and increasing amounts of sulfur in the fluids, culminating with the sulfide-dominated assemblage which appears to close the mineralizing process. The progression from native Bi to Ni–Co arsenide- to base metal sulfide-rich assemblages is observed in several five-element vein-type ore systems described in literature (e.g., [1,8,85] and references therein). Discriminant features between the two vein systems are represented by the nature

of the lithologies hosting and close to the ore systems at regional scale as well as the variable rates of enrichment of Ni (Sardinia) versus Co (Usseglio). The Ni-rich SW Sardinian deposits are hosted in fault systems crosscutting Paleozoic siliciclastic basement rocks and flanking two late Variscan plutonic complexes, while the Co-rich Usseglio veins outline a fracture system crosscutting one of the largest ophiolite complex of the Western Alpine belt. The timing of the mineralization appears to be better constrained for the Usseglio veins, which may be late- to post-Alpine in age [41].

The gabbro-hosted Cu–Ag–Pb–Te–Se stockwork ore at Piazza, in the Eastern Ligurian sector of the Appennine belt, has been described here for the first time to some detail, with its unique assemblage comprising bornite, chalcopyrite, siegenite, millerite, tellurides, selenides, and sphalerite in carbonate-free prehnite–chlorite–albite gangue. In the area siegenite and millerite are not exclusive of Piazza, as they are part of the accessory, secondary Fe–Ni–Cu sulfide assemblage disseminated within calcite stockworks of the ophicalcite quarries near Levanto and Bonassola, south of Piazza (see Figure 11b). But the calcite veining in the quarries, interpreted as consequent to oceanic serpentinization processes of slow/ultra-slow spreading ridges by [51], and the carbonate-free Piazza stockwork ore are not comparable. On the contrary, the Piazza mineralization does broadly share several out of the elemental components of the five element vein systems, i.e., Co, Ni, Ag, base metals, although it shows marked differences from that metallogenic model. The Piazza deposit shares the ophiolite-related geological setting with the Usseglio veins and its seleniferous character with the SW Sardinian veins. But at Piazza the Co–Ni mineralization only occurs in accessory amounts compared to the two other vein systems. Other major differences are represented by the lack or paucity of As and Sb (Co and Ni are fixed in sulfides only) and by the absence of carbonates.

As previously mentioned, the Piazza deposit is included in a regional-scale metallogenic framework (Ligurian Appennines) characterized by remarkable Cyprus-type seafloor/sub-seafloor deposits related to the Jurassic Tethys Ocean. However, we think that the Piazza vein stockwork differs from the other types of ophiolite-hosted volcanogenic Ligurian sulfide deposits so far classified by [48,49] (and ref. therein). No geochronological data are available for Piazza at the moment, although especially the mixed nature of the complex metallic assemblage, with "mafic" Cu, Ni, and Co coupled with the enrichments in Ag and Pb, affine to continental crust, may suggest ore hydrothermal deposition with metal reworking in relation to the Tertiary Appennine orogeny. This hypothesis is in agreement with what supported by [86,87] in reference to the prehnite- and zeolite-rich assemblages and to the enrichments of "continental" components, like Sr and Ba, in the gabbro-hosted stockwork at Campegli. If this is the case, the similarity between Piazza and the Usseglio late- to post-Alpine veins is further enhanced.

In their own regional metallogenic contexts, the Ni-rich SW Sardinian and the Co-rich Usseglio vein systems represent unique examples of five element vein-type mineralization. Southern Sardinia, and especially the SW sector, is extremely rich in various types of ore deposits (see a review in [88]) but is relatively scarce in mafic-ultramafic rocks (i.e., potential Ni and Co sources), represented by rather poorly exposed gabbros in the Variscan Arbus and Capo Pecora intrusive complexes [14]. The Western Alps are, on the contrary, relatively rich in mafic to ultramafic rocks from ophiolite massifs. Also, the Western Alps are host to large-scale, late- to post-metamorphic hydrothermal activity producing mesothermal to epithermal gold deposition during late Tertiary in both gneissic and ophiolitic terranes ([31] and ref. therein). However, the Usseglio vein set (together with the minor Cruvino deposit, in the nearby Susa Valley, [89]) appears to be the only major Co–Ni-rich Alpine hydrothermal pole.

The Northern Appenninic ophiolite massifs of Bracco and Valgraveglia (Internal Liguride complexes) appear to have the greatest potential for hydrothermal Co–Ni mineralization because there are evidences of regionally extended proto-enrichments of Co and Ni related to the widespread volcanogenic sulfide deposits. With their data, [49] confirmed the vocation of the Ligurian copper deposits to contain highly variable but significant Co and Ni concentrations (up to 5 wt% Co and 1.7 wt% Ni) mostly hosted in pyrite [47]. Notably, serpentinite-hosted volcanogenic mineralization

and related Fe sulfides tend to display lower bulk Co/Ni ratios than stratiform ores hosted in basalt and in contact with the pelagic sedimentary units. This suggests a direct contribution of components from the host rocks and, in particular, the affinity of Co for seafloor environments [90,91]. For the sulfide-bearing carbonated serpentinites in the ophicalcite quarries, [51] report the dissemination of accessory Co-rich pentlandite (up to 13.4 wt% Co), Ni-rich pyrite and pyrrhotite crystals (both up to 2 wt% Ni), in addition to millerite and Co–Ni siegenite. With its assemblage the Piazza mineralization appears to embody a further step in a progressive Co and Ni enrichment via metal reworking by hydrothermal fluids, although the process did not develop into either an economic nor a full five element vein-type deposit.

What is discriminating the minor Piazza mineralization from the comparably economic five element vein-type ore systems of Usseglio and SW Sardinia?

The first quantitative determinations of some physico–chemical characters of the ore-forming fluids, including temperature, have come via fluid inclusion analyses only for the Usseglio and for the SW Sardinian vein systems. For the Piazza deposit temperature estimates are derived from the semi-empirical chlorite geothermometer and speculations only can be made for the ore fluids on the basis of the mineral assemblages. The data obtained on the fluid inclusions from the three main hydrothermal stages of the Usseglio vein system, although preliminary, give information on the nature of the fluid and its evolution with time. Microthermometric data indicate that the hydrothermal fluid was characterized, from the earliest to the mineralization stage, by a relatively uniform composition: the inclusions in all stages show a relatively high salinity (17.9 to 24.4 equiv. mass% NaCl) which is due not only to $Na^+$ and $Cl^-$, but also to $Ca^{2+}$ in significant proportion and possibly—at least during the earliest siderite stage—sulfate dissolved in the fluid. Considering the H2O–NaCl–CaCl2 system, the calculation by use of $Tm_{ice}$ and $Tm_{hh}$ pairs according to [65] suggests an average salinity of about 20 mass% $NaCl + CaCl_2$, with a $NaCl/CaCl_2$ mass ratio mostly comprised between 1.6 and 4.5. The $CO_2$ content was instead very low (<ca. 1.5 mol%, as suggested by the absence of clathrate, [92]) and the high siderite content is likely due to its extremely low solubility under hydrothermal conditions. The fluid inclusions petrography shows that an important boiling event occurred in the system. Considering the hydrothermal circulation within the framework of the late alpine brittle evolution [40,41,93], boiling could be explained by a sudden drop in pressure (from supra-hydrostatic to hydrostatic) connected with the brittle structural deformation. Boiling process was likely linked to the strong brecciation that occurred during this stage and triggered precipitation of abundant barite. If boiling occurred, $T_h$ of the liquid-rich fluid inclusions, i.e., ca. 220 °C, can be assumed as the temperature of trapping during the baryte stage. The depth of boiling of a fluid with a salinity of about 20 mass% NaCl and temperature of 220 °C is around 185 m below the water table corresponding to a pressure of 2 MPa [94]. Since the fluid surely contained some $CO_2$ (although in very minor amount), such pressure value must be considered as a minimum estimate. The Co mineralization formed during the subsequent Co–Ni–Fe arsenides stage. The fluid inclusions data show that deposition occurred from a homogeneous, non-boiling fluid. Boiling possibly played an indirect role for the native metals and arsenides deposition (e.g., by removing part of the sulfur from the system?) but was not the ore precipitation mechanism. The $T_h$ value of ca. 170 °C of fluid inclusions related to the Co–Ni–Fe arsenides stage represents a minimum temperature estimate for the cobalt mineralization. However, considering depth of deposition during the baryte stage, we can also assume a similar shallow depth during the subsequent stage of deposition. Under such conditions, pressure correction to the fluid inclusions $T_h$ is negligible and $T_h$ values can be considered near equal to the fluid inclusions trapping temperatures. As a drastic change in the fluid composition is not observed, the relatively sharp temperature decrease recorded ($T_h$ falls from ca. 220 °C of the baryte stage to ca. 170 °C) was probably important for the ore deposition. Such a temperature decrease may be related to the boiling event or to mixing with a lower temperature fluid; however, unequivocal evidence of fluid mixing (that is typically represented by a change in fluid composition) was not found so far.

Regarding the SW Sardinian vein system, fluid inclusion-rich quartz adequate for analysis appears to occur as a late phase in the mineral assemblage of the Co–Ni ores. Hence, the fluid inclusion data provide information on late-stage conditions of deposition, corresponding to base metal sulfide mineralization. The examined fluid inclusions are characterized by low $T_h$ values (mostly between 60 and 100 °C), relatively high salinity (from 20.1 to 24.8 equiv. mass% NaCl) and the possible occurrence of dissolved $Ca^{2+}$ in fluid, addition to $Na^+$ and $Cl^-$. Such features are similar to those of fluid inclusions from the nearby Montevecchio Pb–Zn–Ag deposit studied by [30]. In particular, several data of our fluid inclusions fall within or close to the fields of fluid inclusions in quartz from Montevecchio, suggesting a genetic link of the fluids that circulated in the two vein systems located at relatively short distance from one another. Moreover, the $T_h$ and salinity range and the NaCl–CaCl2 composition of fluid inclusions of the Ni–Co-bearing mineralization from SW Sardinia are comparable with those trapped during the Jurassic-Cretaceous event of the multi-stage Ag–Bi–Co–Ni–U ore deposits of Wittichen (SW Germany; [95]). More in general the fluid inclusion features of SW Sardinia are comparable with those of the fluid trapped in inclusions of ore and gangue minerals in post-Variscan (i.e., Jurassic-Cretaceous) Ba-F-(Pb–Zn–Ag) and Pb-Zn. Ag deposits in Sardinia and Central-Western Europe [96–102]. From these analogies we can assume that at least the late depositional phase of the Co–Ni mineralization of SW Sardinia can be related to the circulation of a post-Variscan, Triassic to Jurassic-Cretaceous low-temperature H2O–NaCl–CaCl2 brine. According to [98] the origin of this type of fluid responsible for the genesis of some post-Variscan mineralization in Sardinia is mainly from evaporated seawater, with a possible small contribution from halite dissolution. Post-Variscan brine associated to the deposition of dolomite in Zn–Pb–Cu veins at the Nızky Jesenık Mountains and the Upper Silesian Basin (Czech Republic) are also interpreted as evaporated sea-water [103]. Whereas, [96] suggested that the mixing of post-Variscan NaCl–CaCl2-bearing basinal brines with low-temperature sulphate-rich formation waters can explain the temperature/salinity features of fluid inclusions from post-Variscan mineralization of the Harz (Germany) mining district. Since no evidence of boiling was noted in from fluid inclusion inspection, $T_h$ of LV inclusions are a minimum estimate of fluid temperature at trapping. Thus, in order to compute the trapping temperature an estimate of the depth and consequently of pressure of the fluid flow is needed. However, the depth of the Ni–Co mineralization of SW Sardinia in post-Variscan times is difficult to constrain from the available geological data. In general, a shallow formation depth was proposed for Jurassic-Cretaceous deposits related to the flow of NaCl–CaCl2 brines. For example, a maximum cover of about 1500 m was considered for the Wittichen mineralization (SW Germany) by [95], whereas a thickness of 2–3 km the eroded cover was estimated for the hydrothermal vein-style mineralization of Cretaceous age of Freiberg (Germany) by [102]. Thus, assuming that also the Ni–Co mineralization of SW Sardinia were formed at similar shallow depth for a maximum thickness of cover of 3 km, corresponding to a maximum hydrostatic pressure of 30 MPa and a lithostatic pressure of 76 MPa (considering an average rock density of 2.6 g/cm$^3$). For such pressure values, the pressure corrections to add to the $T_h$ values for trapping temperature computation are 10 and 25 °C, respectively [104]. At relatively shallow depth hydrostatic pressure conditions predominate, although transient pressure above hydrostatic pressure can occur as a consequence of self-sealing process. Thus, trapping temperature for the late-stage ore must be in general comprised between 52 and 126 °C.

Considering the temperature interval determined from fluid inclusion analyses, we derived an approximate oxygen isotope composition of the fluid in equilibrium with the carbonates in the SW Sardinian and Usseglio veins via experimental siderite-water fractionation equations by [105–107] selected through the stable isotope calculator facility by [108,109]. The resulting $\delta^{18}O$ estimate of the fluid in the temperature interval between 125 °C and 200 °C is between 0.0–1.7‰ and 5.3–5.7‰ for the SW Sardinian veins and between 2–3.69‰ and 7.3–7.7‰ for the Usseglio veins. These values partly overlap with the restricted oxygen isotope composition of the magmatic fluids (+5.5–+10‰; [110]) but are also included in the widely variable compositional range for oxygen isotopes determined for ore-related brines and, in general, basinal-oil-field brines and formation waters ([82,111] and references

therein). The evaluation is speculative because no δD measures on fluid inclusions are available for a complete characterization of the fluid. However, the $\delta^{18}O_{fluid}$ values for both SW Sardinian and Usseglio veins are comparable with the wide range of $\delta^{18}O_{fluid}$ signatures (0.47–9.12‰) determined by [82] on quartz and carbonates for the Great Bear Lake ore deposits.

The chlorite-based temperature range of 200–280 °C estimated for the mineralizing process at Piazza (Figure 14c) is within the range of temperatures recorded in five element vein-type deposits (e.g., [1,7,8,76], and references therein). The occurrence of Ag–Pb tellurides and selenides in the Cu ore is compatible with the phase assemblages calculated by [57] at 300 °C (Figure 17a). The diagram in Figure 17b employs siegenite as a surrogate for the buffering Fe phase (pyrite, pyrrhotite, absent at Piazza) for the sulfur fugacity, according to [109,110]. At lower temperature conditions (e.g., at 250 °C, in [112–114]) phase relations for both sulfides and tellurides vary only limitedly. On the basis of the issues discussed in [115], the Piazza assemblage may correspond to an early Te–Se-rich, S-bearing mineralizing stage including Co and Ni sulfides and progressively evolving to S-richer, thereby stabilizing chalcopyrite (plus Cd-rich sphalerite and greenockite) over bornite. Even if no arsenic is involved, such polyphasic depositional scheme, involving a progressive increase of $fS_2$ in the mineralizing fluid, is similar to the trend recorded in five element vein-type mineralization. Piazza might be a variant of the typical As-rich five element vein-type scheme, in a similar way as the Co–Ni siegenite-bearing base metal sulfide veins are in the Siegerland district in Germany [116].

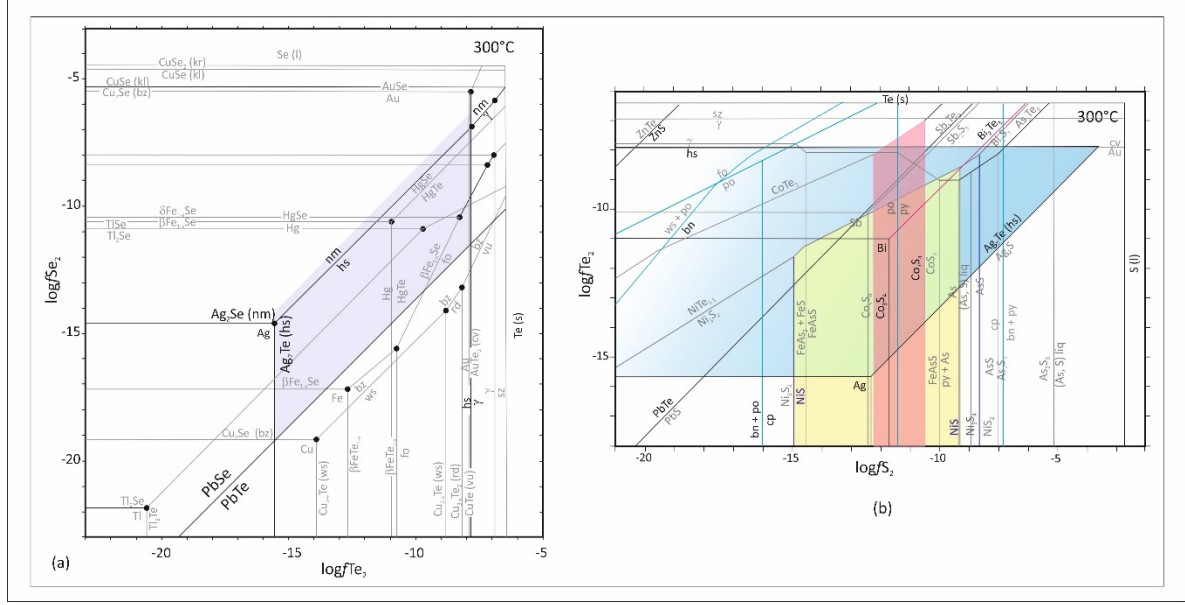

**Figure 17.** Phase relations for selenides and tellurides (**a**) and for selenides, tellurides and sulfides (**b**) in function of log $fTe_2$, $fSe_2$ and $fS_2$ at 300 °C by [57,112–115]. The colored areas represent stability fields for the selenide, telluride and Co–Ni sulfide phases observed in the Piazza mineralization.

Another major difference with five element-type deposits is, however, indicated in the absence of carbon-bearing gaseous components in the fluid, as suggested by the absence of carbonates in the prehnite-rich assemblage of the Piazza mineralization. According to [116–118], formation of hydrothermal prehnite (together with feldspar) would occur in presence of alkali chloride waters hotter than 200 °C and is favored in waters of neutral to slightly alkaline pH and low in dissolved $CO_2$, usually lost through degassing/boiling. Hence, the occurrence of prehnite in a hydrothermal context should primarily indicate that degassing of the hydrothermal fluid in $CO_2$ occurred prior to deposition.

The SW Sardinian and Usseglio vein systems closely conform to recent geochemical models [6,8]. The textural relationships observed in the arsenide phases, such as the depositional sequence of mono- to diarsenides (e.g., nickelite to rammelsbergite, safflorite and löllingite), the stabilization of

skutterudite and the stability of native Bi, are compatible with the conditions outlined in the pH-$f$O$_2$-T °C diagrams in [8]. The authors claim that Ni, Co and Fe availability and oxygen fugacity, more than temperature and variation in as contents, contribute to explaining what observed in natural ores. In the model diagrams, arsenides precipitate between 200° and 250 °C in response to mild fluctuations in oxygen fugacity within a 4–7 pH range. The early native arsenic recorded in Usseglio (Figure 3b) suggests relatively low initial pH and $f$O$_2$ conditions. The predominance, during the early arsenide stage, of Co triarsenide skutterudite, like in the Usseglio mineralization, would also be compatible with relatively lower pH and higher $f$O$_2$ than the Co diarsenide safflorite. Nickel, cobalt and iron display different sensitivity to $f$O$_2$ conditions: the tendency to progressive Co and Fe enrichment relative to Ni, recorded for the mono- to diarsenides in the SW Sardinian paragenetic scheme, may have recorded progressive decrease in $f$O$_2$ from Ni to Co to Fe.

In the relevant literature carbonates occur as essential gangue components in the five-element vein-type deposits and correlate with fluid inclusions carrying a wealth of carbon components including hydrocarbons like methane e.g., [6,7]. According to [6], the sources of methane may be various, although the most common sources may be carbonaceous schists and shales, which are often occurring in the sequences hosting the ore deposits. No data are, for the moment, available about possible fluid inclusion-hosted hydrocarbon components in the ore deposits presented here. But there are several evidences of interaction of fluids with such rock types for all the deposits considered, including Piazza. The SW Sardinian veins do bear direct evidence of the involvement of carbonaceous shales in the mineralization as Silurian black shale units are part of the local sedimentary sequence, and black shale fragments are common in the brecciated ores (e.g., Figure 2b,d,f). An additional indication is provided by the isotopic signatures of the ore-related carbonate gangue. The tendency towards progressively negative $\delta^{13}$C signatures is considered as reflecting a role of organic carbon deriving from interaction of the fluids with carbonaceous shales and/or from the presence of hydrocarbon components in the migrating ore brines (e.g., [1,6,7]). The same applies also to the carbonate gangue from Usseglio, that shows even more negative $\delta^{13}$C signatures than most of the SW Sardinian carbonates (Figure 16a,b). An indirect evidence may be envisaged in the occurrence/abundance of selenium in the ore minerals recorded both in the SW Sardinian veins and, especially, in the Piazza deposit. Selenium is known to be enriched in the Silurian sulfur-rich carbonaceous, black shales in Sardinia [119]. The selenium enrichment in the Piazza mineralization might have derived from the interaction of the fluids with the numerous outcrops of pelagic shales in the Bracco ophiolite massif (see Figure 11b). However, the absence of carbonates at Piazza seems to coincide with a "failed" Co–Ni ore deposit in a high-potential setting.

The Usseglio vein system shares with Bou Azzer several similarities, particularly the ophiolite-related geological setting, the marked Co enrichment in ore and the evolution of the di- and sulfarsenides, that plots them in the Ni–Co-Fe arsenide-dominated ores subclass in [8]. The Sardinian veins display, instead, a higher Ni/Fe ratio; a suitable source of Ni and Co are mafic intrusives from the nearby Arbus pluton, leached by fluids of undetermined origin. The reason why both Co- (e.g., Usseglio) and Ni-rich (e.g., Sardinian veins) systems occur is still an open question. For both Usseglio and Bou Azzer, the Co and Ni source is obviously the ophiolite host, which is strongly enriched in both metals. The Bou Azzer district includes, however, both Co-and Ni-rich ores [82,120,121], suggesting that –at least in the ophiolite-hosted Co–Ni arsenide deposits- the fluid characters, more than the source rock, play a role in the Co/Ni ratio within individual deposits, in agreement with [9]. Since the Co-arsenide fields predominate at lower $f$O$_2$ than those of the Ni-rich systems [8], more reducing hydrothermal systems (i.e., whose fluids interacted more deeply with organic matter, hydrocarbons, or otherwise fluids connected with serpentinization processes, etc.) are likely to produce Co-richer deposits.

## 8. Conclusions

The data presented here, for the first time, on three poorly known Italian Ni- and Co-bearing deposits underline a series of similarities and differences, that variably link these deposits to one another as well as to the metallogenic model of the five-element vein-type mineralization.

The Ni–Co-rich vein systems of SW Sardinia and Usseglio (Western Alps) show strong similarities: in spite of the different tectonic setting (SW Sardinia: Paleozoic basement and late Variscan plutonic complexes, Usseglio: metaophiolite unit), these vein systems show comparable mineral assemblages, characterized by a Ni- or Co-dominated, arsenide-rich stage followed by the deposition of base metal sulfides + tetrahedrite. In both systems gangue is composed of carbonates and quartz (± barite); Ag, Bi and Sb also occur in variable amounts. The progression from native metals (Bi) to Ni–Co arsenide- to base metal sulfide-rich assemblages observed in the Sardinian and Usseglio veins is one of the key features characterizing several five-element vein-type ore systems in different tectonic settings described in literature.

The Piazza gabbro-hosted Cu–Ag deposit in the Northern Apennine belt in Eastern Liguria is located in an ophiolite complex endowed with abundant volcanogenic sulfide deposits anomalous for Co and Ni. The mineral assemblage at Piazza is more complex than reported in previous rare studies, and includes Cu–Fe and Co–Ni sulfides with Ag-and Pb-bearing tellurides and selenides and Cd-rich sphalerite in a prehnite-rich gangue devoid of carbonates. Even if arsenides are lacking, Piazza records a polyphasic depositional scheme which involves a progressive increase of $fS_2$ in the mineralizing fluid similar to the trend recorded in five element vein-type mineralization. Piazza might represent a variant of the As-rich five element vein-type scheme, although this deposit did not develop into economic for Co–Ni in spite of the highly favorable source rocks at regional scale.

Fluid inclusions data at Usseglio show that the hydrothermal fluid was characterized, during its multistage evolution, by a relatively uniform composition, characterized by relatively high salinity, ca. 20 equiv. mass% NaCl + CaCl$_2$, and probably—at least during the earliest stage—by dissolved sulfate. The ore deposition occurred at shallow depth immediately after a strong boiling event at about 220 °C, that triggered baryte deposition and possibly played a role in the mineralizing process, which deposited native metals (As, Bi) and Co–Ni–Fe arsenides from a homogeneous, non-boiling fluid, at about 170 °C. Drastic cooling was probably a main factor for the mineralization. For the SW Sardinian vein system, fluid inclusions data indicates that the fluid associated to quartz deposition during the late evolution of the Ni–Co hydrothermal system is also characterized by NaCl–CaCl2-bearing brine with salinities from 20.1 and 24.8 equiv. mass% NaCl and temperature between 52 and 126 °C considering hydrostatic condition. Hence, in spite of the differences in geological setting, both the Usseglio and SW Sardinian vein systems are characterized by the occurrence of NaCl–CaCl2-bearing brines similar to most five element-vein systems. The origin of the hydrothermal fluids in the individual deposits is not among the objectives of this study. However, the fluid responsible for late-stage deposition of the Ni–Co mineralization in SW Sardinia likely belongs to the basinal brines and formation waters related to the deposition of Pb–Zn–Ag–Cu(–Co–Ni) post-Variscan ores in Sardinia and Central and Western Europe. The origin of the fluids for the Usseglio vein system is less obvious. As, in general, fluids inherit a salinity (though modified through subsequent fluid-rock interaction) that reflects their source and origin [122,123], a fluid source external to the ophiolitic rocks is required: as a working hypothesis this could be represented by formation waters of (meta-)sedimentary sequences lying below the ophiolite unit. Fluid inclusion data are not available for the Piazza deposit. However, a temperature range of 200–280 °C has been estimated for the mineralizing process, based on chlorite geothermometry and phase assemblages. Such temperature conditions are compatible with the five-element vein-type model.

The complex textural relationships among arsenide phases, sulfarsenides and native Bi in the SW Sardinian and Usseglio veins are consistent with observations from other five elements systems and can be explained by fluctuations in oxygen fugacity within a 4–7 pH range, as modeled in [8]. According to these Authors, a reducing fluid for deposition of the arsenide assemblages is required, which is often documented by the occurrence of carbon components, including hydrocarbons like methane,

e.g., [6,7]. Although the presence of hydrocarbons has not been documented so far in the studied systems, evidences of interactions between hydrothermal fluids and carbonaceous material are given by the occurrence of black shale fragments in the brecciated ores (SW Sardinia) and by the markedly negative $\delta^{13}$C signature of the carbonate gangue (SW Sardinia and Usseglio). An additional, indirect evidence may be envisaged in the occurrence of selenium (from black shales) in the ore minerals both in the SW Sardinian veins and, especially, in the Piazza deposit. The studied systems range from Co-(e.g., Usseglio) to Ni-rich (e.g., Sardinian veins). Multiple factors control the Co/Ni ratio that can be strongly variable, even within the same deposit, during the mineralizing episodes. Although the fluid source is obviously important, the fluid characteristics play an important role in the Co/Ni ratio within individual deposits. Since the Co-arsenide fields predominate at lower $f$O$_2$ than those of the Ni-rich systems [8], more reducing hydrothermal systems (i.e., whose fluids interacted more deeply with organic matter, hydrocarbons, but possibly also fluids connected with serpentinization processes) are likely to produce Co-richer deposits.

**Supplementary Materials:** The following are available online at http://www.mdpi.com/2075-163X/9/7/429/s1, Supplementary 1—Analytical methods and instruments. Table S1—Major element chemistry of carbonates, arsenides and base metal sulfides from the Southern Arburese/SW sardinian vein system. Table S2—Major element chemistry of carbonates, arsenides and base metal sulfides from the Usseglio vein system. Table S3—Chemical analyses for sulfides, sulfosalts and gangue minerals (chlorite, prehnite, albite) from the Piazza stockwork. Table S4—Microthermometric measurements for fluid inclusions from Usseglio and SW Sardinian vein systems. Table S5—Carbon and oxygen isotope composition of Fe carbonates from selected samples of the SW Sardinian/Southern Arburese (a) and the Usseglio (b) vein systems.

**Author Contributions:** M.M., S.N. and P.R. conceived the paper and lead the work; S.N., L.M., M.M., G.O. and F.S. worked on the Sardinian deposits; P.R. and D.C. worked on the Usseglio deposits; fluid inclusion analyses were performed by P.R. on the Usseglio samples and by G.R. and A.A. on the Sardinian samples. E.F. and M.M. performed stable isotope analyses; A.F., M.M. and P.T. worked on the Piazza deposit.

**Funding:** This research and the APC were funded by research grants from the Italian Government (MIUR) and by FdS-Unica 2018 (PI G.B. De Giudici), which is gratefully acknowledged by S. Naitza.

**Acknowledgments:** S. Naitza and M. Moroni would like to thank Proff. Pierfranco Lattanzi and Ida Pirri Venerandi for long and fruitful discussions and exchange of ideas about metallogeny in SW Sardinia, P. Rossetti is indebted to Prof. Stefano Zucchetti for having introduced him to the fascinating world of hydrothermal ore deposits. M. Moroni, L. Magnani and P. Tartarotti are indebted to Fabio Marchesini and Andrea Risplendente (Earth Science Department, State University of Milano) for their professional help in sample preparation and for the assistance during the sessions of electron microprobe analysis. M. Moroni, P. Tartarotti and A. Franklin would like to thank Giovanni Grieco for introducing them to the Piazza mine site. The authors would like to thank the personnel of MDPI for the editorial handling and the reviewers for the advices and the help in the revision of the manuscript.

**Conflicts of Interest:** The authors declare no conflicts of interest.

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
