# Peer review of "Factors Controlling Hydrothermal Nickel and Cobalt Mineralization—Some Suggestions from Historical Ore Deposits in Italy"

_minerals, doi:10.3390/min9070429_

Round 1

Reviewer 1 Report

Well written article. Just be careful using the same name for the mineral allover the text.

The authors describe in this article a few small historical Ni-Co-bearing deposits from Italy and present their mineralogical, chemical, thermal, and stable isotope characteristics. They suggest, and this is true, the deposits they describe share many similarities that overlap the Five-Element-type of deposits from elsewhere. The authors significant contributions are not related only to the very detailed mineral description, that is really very attractive, but they make considerations on the thermal conditions of depositions. Using the fluid inclusions study, the authors bracket the origin of the ore fluids as being derived from the basin brines. They discus, based on geochemical data, the source-rock and its control on the Ni-Co mineralization and the possibility for discovery of an economic ore body. The authors presented the characteristics of small deposits but their data and interpretation are completing those obtained from the classical deposits

Author Response

Dear Sir/Madam,

thank you for reviewing our manuscript and for appreciating our effort. We considered your indications in the revision we are submitting.

Thank you and best regards

Marilena Moroni and co-authors

Reviewer 2 Report

The manuscript concerns interesting and till present insufficiently investigated ore mineralization, which is worth to study both from scientific and practical points of view. The scheme of the text is correct and investigation approach presented properly. The set of the methods applied yielded results which are a good basis of the genetic problems discussion of the studied ore mineralization and for the conclusions presented by the authors.

However, some improvements are necessary. Generally, the text at places is too extensive, written with unnecessary details. I evaluate, that the manuscript could be shortened (10-15%) by use more concise, thus frequently clearer style. Some statements are not logical, as indicated in the sticky notes. Especially ambiguous is use of slash, which has unclear meaning, except for the cases, when it indicates ratio of two values (but in the 17th and 18th centuries slash was used as comma). Use of the not precise and poorly metaphoric phrases makes impression of the jargon (“chemistry” for chemical properties, “geology” for geological features etc.)

The calculated average values have not scientific importance and should be deleted. They would suggest that the ranges of the measured values indicate the size of the laboratory error, NOT concerning the changes of the salinity or temperature during formation of the minerals studied. Only in the case of error ranges information the average values would have real meaning.

The group of the “pseudosecondary” (in other texts also called “primary-secondary”) fluid inclusions was established by N. P. Ermakov to increase his scientific achievements. However, the name „pseudo-secondary” (as well as “primary-secondary) is a scientific (and logical) oxymoron. The definition is, that such inclusions formed as secondary in a crystal from the fluid, which was the parent one for a growth zone of the same crystal. However, it is not possible to find in a preparation plate ALL growth zones of the studied crystal, thus it is not possible to recognize as well, if a secondary inclusion is only secondary or “pseudosecondary”. Also in quartz crystal crack in its flat section may have shape of wedge or a part of ellipse starting from a face of quartz and in some orientations of the preparation it does not reach other faces of the crystal. Moreover, even recognition of the “pseudosecondary” inclusions does not help in genetic interpretation – much more important is to arrange the sequence of the secondary inclusion generations e. g. by observations of the inclusion refilling phenomena.

Early report on the CaCl2-rich inclusion solution was published by A. Kozłowski (Calcium-rich inclusion solutions in fluorite from the Strzegom pegmatites, Lower Silesia, Acta Geologica Polonica 34 (1-2), 131-137, 1984, see in Research Gate), who evidenced presence of this compound in the solution by observations of the freezing phenomena and the refractive indices of the inclusion solutions.

The text contains some linguistic errors, like the phrase “based on”, otherwise common in various publications, whereas correctly should be written “on the basis of”, see T. Neil Irvine and Douglas Rumble III, 1992, A writing guide for petrological (and other geological) manuscripts, p. 11; Journal of Petrology, v. 33, no. 4, suppl. Passive voice linked with present participle is another error (e. g. “was made using”), ditto, p. 5. These and other mistakes are marked in the text. The manuscript should be thoroughly considered from the linguistic point of view.

My individual remarks may be found in the sticky notes in the manuscript text.

Author Response

Dear Sir,

Thank you for the careful reading and revision of the manuscript and for the many suggestions for improving it. We are submitting revised files, related to the manuscript, in which we fixed all, or almost all the mistakes and improper terms. 

We apologize for the use and abuse of slash, which was meant to be erased before the first submission but which was forgotten.

Regarding the length of the manuscript, we are aware it is rather long. During the revision we tried to shorten the text by trimming paragraphs. The difficulty in properly arranging the format of the files (with the shift of figures) is not allowing me to evaluate the extent of the shortening of the text. Surely, we did not manage in shortening as much as the 15% required by the reviewer.

However, the length is at least partly justified by the fact that we have been dealing with three different deposits which are basically unknown and with little chances of referring to recent papers for details. The choice of dealing with these poorly known deposits had been discussed with the editor of the special edition on high-tech resources, as one of the goals of the edition (in the description of the special issue) was to gather information about little explored resources. The sections describing various aspects of these three deposits are actually not very extensive and the figures have been kept to a limited number. Moreover, a contribution to the length also comes from the extensive bibliographic references, related to the different geological settings and topics involved. Anyway, we tried to reduce the text to some extent and we hope that at least the quality will be better than before.

Reviewer comment:

“The calculated average values have not scientific importance and should be deleted. They would suggest that the ranges of the measured values indicate the size of the laboratory error, NOT concerning the changes of the salinity or temperature during formation of the minerals studied. Only in the case of error ranges information the average values would have real meaning.”

 Answer: We accepted the suggestion of the reviewer and we eliminate average values of microthermometric data in the text and in Table 1 

Reviewer comment:

“The group of the “pseudo-secondary” (in other texts also called “primary-secondary”) fluid inclusions was established by N. P. Ermakov to increase his scientific achievements. However, the name „pseudo-secondary” (as well as “primary-secondary) is a scientific (and logical) oxymoron. The definition is, that such inclusions formed as secondary in a crystal from the fluid, which was the parent one for a growth zone of the same crystal. However, it is not possible to find in a preparation plate ALL growth zones of the studied crystal, thus it is not possible to recognize as well, if a secondary inclusion is only secondary or “pseudo-secondary”. Also in quartz crystal crack in its flat section may have shape of wedge or a part of ellipse starting from a face of quartz and in some orientations of the preparation it does not reach other faces of the crystal. Moreover, even recognition of the “pseudo-secondary” inclusions does not help in genetic interpretation – much more important is to arrange the sequence of the secondary inclusion generations e. g. by observations of the inclusion refilling phenomena.

 Answer:

We accepted the suggestion of the reviewer, since we cannot distinguish secondary and pseudo-secondary inclusions we consider fluid inclusions along healed fractures as secondary or pseudo-secondary. 

Reviewer comment:

“Early report on the CaCl2-rich inclusion solution was published by A. Kozłowski (Calcium-rich inclusion solutions in fluorite from the Strzegom pegmatites, Lower Silesia, Acta Geologica Polonica 34 (1-2), 131-137, 1984, see in Research Gate), who evidenced presence of this compound in the solution by observations of the freezing phenomena and the refractive indices of the inclusion solutions.”

 Answer:

The behaviour of fluid inclusions during low-temperature experiments from Sardinia is similar to that illustrated by A. Kozłowski thus we add this reference in the section 5.2.1 of the manuscript and as [66] in the reference list.

Here are additional answers to comments by the reviewer:

Line 30 (abstract) – The geothermometer does not provide the indication about the CO2 in the fluid. The sentence in the abstract was compressed because of the space limitation and the result of this was unfortunately weird. Thank you for pointing it out.

Line 63 - We do not fully understand the objection here and in the other places in the manuscript where this note is found. Indeed, the term “Appennines” corresponds to the Italian phrase Monti Appennini, but in geological papers in international journals the term Appennines is widely used instead of the Italian name. However, in the text we changed it into Appennine Belt.

Line 71 (and other notes) - The phrase "mineral chemistry" is frequently used in many papers for indicating "chemical composition of minerals" in a short way. This is the reason for which we used this expression. However, we modified the text according to the request of the reviewer.

Line 75 – We meant that geological departments often own rock and mineral collections (also ore minerals from mining areas) which date back long time and are cured in a similar way as museum collections.

Line 719 – Pennine term rejected by IMA. Yes, we know about this. Actually, there are plenty of papers, even published very recently which use the classification diagram by Hey (1954), in which the Pennine/Penninite field is reported. I happened to see only one paper (Wu et al., 2019, in Minerals) applying the classification by Zane and Weiss (1998). We decided to leave the classification diagram and to eliminate the term “pennine” from the text, in the same way as seen in recent papers. We hope the modification is acceptable.

Thank you again for your help and best regards

Marilena Moroni (on behalf of the co-authors)